# Early Decisions Matter: Proximity Bias and Initial Trajectory Shaping in Non-Autoregressive Diffusion Language Models

Jiyeon Kim [1] [*]   Sungik Choi [2]   Yongrae Jo [2]   Moontae Lee [2] [3]   Minjoon Seo [1]

## Abstract

Diffusion-based language models (dLLMs) have emerged as a promising alternative to autoregressive language models, offering the potential for parallel token generation and bidirectional context modeling. However, harnessing this flexibility for fully non-autoregressive decoding remains an open question, particularly for reasoning and planning tasks. In this work, we investigate non-autoregressive decoding in dLLMs by systematically analyzing its inference dynamics along the temporal axis. Specifically, we uncover an inherent failure mode in confidence-based non-autoregressive generation stemming from a strong *proximity bias*—the tendency for the denoising order to concentrate on spatially adjacent tokens. This local dependency leads to spatial error propagation, rendering the entire trajectory critically contingent on the initial unmasking position. Leveraging this insight, we present a minimal-intervention approach that guides early token selection, employing a lightweight planner and end-of-sequence temperature annealing. We thoroughly evaluate our method on various reasoning and planning tasks and observe substantial overall improvement over existing heuristic baselines without significant computational overhead.

## 1. Introduction

Autoregressive Large Language Models (LLMs) have demonstrated remarkable success in various tasks (Brown et al., 2020; Touvron et al., 2023; Achiam et al., 2023), but their strictly sequential nature introduces a fundamental bottleneck. On the other hand, Diffusion-based Large Language Models (dLLMs) (Ou et al., 2024; Sahoo et al., 2024; Shi et al., 2024; Nie et al., 2025; Ye et al., 2025b) offer a conceptually appealing alternative, promising two distinct advantages: parallelism and bidirectionality. Unlike autoregressive LLMs that generate tokens sequentially conditioned on previous context, dLLMs can generate multiple tokens simultaneously (parallelism) and iteratively refine the sequence, leveraging global context from both directions (bidirectionality).

Despite these theoretical advantages, fully non-autoregressive (NAR) decoding has struggled to achieve competitive performance in practice, often suffering from incoherent generation (Seo et al., 2025). To mitigate this instability, current state-of-the-art methods typically resort to semi-autoregressive decoding (Arriola et al., 2025; Nie et al., 2025; Wei et al., 2025), which injects structural priors by generating blocks of tokens sequentially. While effective, this reliance on semi-autoregressive heuristics imposes a critical trade-off that may undermine the core benefits of dLLMs. First, reintroducing sequential dependencies inevitably creates a latency bottleneck, limiting the inference speedup gained from parallel decoding (Seo et al., 2025; Wu et al., 2025b; Kim et al., 2025b). Second, imposing a sequential order may restrict the potential of dLLMs for complex reasoning and planning capabilities (Ye et al., 2025a). As a consequence, semi-autoregressive decoding functions as a practical workaround rather than a fundamental resolution to the instability of fully non-autoregressive diffusion.

This reliance on semi-autoregressive decoding leaves a critical question unanswered: *What core failure mechanisms hinder the viability of fully non-autoregressive decoding?* Despite the growing body of work on sampling methods of dLLMs, there remains a limited empirical understanding of how dLLMs behave under fully NAR decoding. In particular, little is known about how the model transitions from a fully masked state to a coherent sequence, and how sensitive this trajectory is to the initial denoising decisions.

To bridge this gap, we revisit non-autoregressive decoding not as a regime to be avoided, but as a distinct setting whose failure modes can be explicitly characterized and resolved. Through empirical analysis, we identify a phenomenon we term **proximity bias**—an intrinsic tendency where the model

---
[*]Work done during an internship at LG AI Research. [1]KAIST AI [2]LG AI Research [3]University of Illinois Chicago. Correspondence to: Jiyeon Kim <jiyeon.kim@kaist.ac.kr>.

*Proceedings of the 43rd International Conference on Machine Learning*, Seoul, South Korea. PMLR 306, 2026. Copyright 2026 by the author(s).

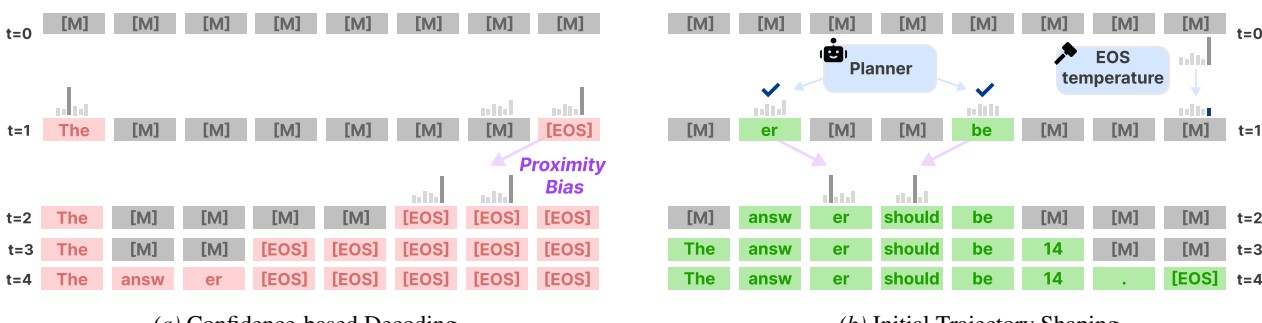

*Figure 1.* (a) Confidence-based sampling in non-autoregressive decoding suffers from premature constriction of the generation window due to *proximity bias*, where neighboring tokens accumulate confidence sequentially. This leads to spatial error propagation, making early unmasking decisions decisive for the entire trajectory. (b) We introduce a lightweight planner and EOS temperature annealing to strategically guide initial position selections toward the target trajectory

rapidly accumulates confidence based on local token associations rather than global coherence. We demonstrate that this characteristic becomes particularly detrimental in confidence-driven non-autoregressive decoding, as visualized in Figure 1a. Unlike semi-autoregressive methods that enforce structural constraints, unrestricted confidence-based sampling causes the model to default to generic statistical priors under high initial uncertainty, which is irreversibly propagated throughout the subsequent generation. Crucially, we find that proximity bias exacerbates premature End-of-Sequence (EOS) prediction, severely truncating the generation window required for reasoning, a phenomenon recently characterized as *EOS overflow* in Kim et al. (2026). Building on these observations, we reveal a critical **temporal asymmetry of importance**: the validity of the final output is disproportionately determined by the unmasking decisions made in the very first denoising step.

Leveraging these insights, we introduce two strategic interventions in sampling order, focused on the initial decoding step, as illustrated in Figure 1b. First, we propose a lightweight planner trained to predict the optimal set of initial positions, anchoring the trajectory toward the target generation. Second, to counteract the tendency toward premature constriction of the generation window, we apply EOS temperature annealing, which lowers the unmasking priority of EOS tokens during the initial phase. Remarkably, even when paired with standard greedy decoding for token selection, these minimal interventions in position selection exhibit significant performance gains on reasoning-intensive tasks. Furthermore, our method effectively generalizes to higher-step regimes in a plug-and-play fashion, highlighting the broad applicability of our design.

Our contributions can be summarized as follows.

- We identify proximity bias and premature EOS dominance as a major reason of primary failure mode in non-autoregressive decoding and reveal a temporal asymmetry

where the initial position selection disproportionately dictates the reasoning trajectory.

- We propose a lightweight planner and an EOS annealing strategy that guide early inference steps without fine-tuning the diffusion backbone.

- Our approach improves performance across low- and high-compute regimes with negligible overhead, while remaining compatible with standard sampling heuristics.

## 2. Preliminaries

### 2.1. Masked Diffusion Language Models

We briefly review the framework of Masked Diffusion Language Models (MDLM) (Sahoo et al., 2024; Shi et al., 2024; Ou et al., 2024); for a detailed mathematical formulation, see Appendix A. MDLM is defined by a forward process that progressively corrupts clean data $\mathbf{x}$ into a masked state via the following marginal and transition distributions:

$$q(\mathbf{z}_t|\mathbf{x}) = \mathrm{Cat}(\mathbf{z}_t; \alpha_t \mathbf{x} + (1 - \alpha_t)\mathbf{m})$$

$$q(\mathbf{z}_s|\mathbf{z}_t, \mathbf{x}) = \begin{cases} \mathrm{Cat}(\mathbf{z}_s; \mathbf{z}_t) & \text{if } \mathbf{z}_t \neq \mathbf{m} \\ \mathrm{Cat}(\mathbf{z}_s; \frac{(1-\alpha_s)\mathbf{m} + (\alpha_s - \alpha_t)\mathbf{x}}{1 - \alpha_t}) & \text{if } \mathbf{z}_t = \mathbf{m} \end{cases}$$

where $\alpha_t \in [0, 1]$ is predefined noise schedule and $\mathbf{m}$ represents the [MASK] token vector.

The reverse process is approximated by a neural network via parameterized posterior $p_\theta(\mathbf{z}_s|\mathbf{z}_t) := q(\mathbf{z}_s|\mathbf{z}_t, \hat{\mathbf{x}}_\theta(\mathbf{z}_t, t))$. The learning objective is to minimize the Negative Evidence Lower Bound (NELBO), defined as the weighted integral of the log-likelihood for clean data $\mathbf{x}$ with $L$ tokens:

$$\mathcal{L} = \mathbb{E}_q \left[ \int_0^1 \frac{\alpha_t'}{1 - \alpha_t} \sum_{l=1}^L \mathbb{1}\left[\mathbf{z}_t^l = \mathbf{m}\right] \left(\mathbf{x}^l \cdot \log \hat{\mathbf{x}}_\theta^l(\mathbf{z}_t, t)\right) \mathrm{d}t \right]$$

where $\alpha_t'$ denotes the time derivative of $\alpha_t$. The expectation is taken over $\mathbf{x} \sim q_0$ and $\mathbf{z}_t \sim q_t(\mathbf{z}_t|\mathbf{x})$.

## 2.2. Inference and Unmasking Strategy

During inference, the reverse process generates data by iteratively denoising a sequence starting from fully masked tokens, $\mathbf{z}_{t_T} = \mathbf{z}^{1:L}$. The continuous diffusion time $t \in [0, 1]$ is discretized over $T$ finite steps, where $1 = t_T > t_{T-1} > \cdots > t_0 = 0$.[1] At each step, the subsequent latent $\mathbf{z}_{t_{i-1}}^{1:L}$ is sampled from the current state backward:

$$\mathbf{z}_{t_{i-1}} \sim p_\theta(\mathbf{z}_{t_{i-1}} | \mathbf{z}_{t_i}), \quad i = T, \cdots, 1$$

In simplified MDLM (Sahoo et al., 2024), the process is characterized by its *absorbing nature*: once a token is unmasked, it remains unchanged in subsequent steps. Consequently, the transition at each step involves two simultaneous decisions: *Token prediction* for currently masked positions $\mathcal{M}_d = \{l \mid \mathbf{z}_d^l = \mathbf{m}\}$, and *Position selection* $\mathcal{U}_d \subseteq \mathcal{M}_d$ to be unmasked at timestep $d$.[2]

We formally define a generation trajectory $\tau$ as the sequence of latent states $\tau = (\mathbf{z}_{t_T}, \mathbf{z}_{t_{T-1}}, \cdots, \mathbf{z}_{t_0})$. While $\mathcal{U}_d$ can theoretically be chosen with random sampling, a confidence-based heuristic is typically employed, where the tokens with the highest model confidence are prioritized for unmasking (Chang et al., 2022). In this work, we focus on this position selection strategy, investigating how decisions made in the early inference steps shape the entire generation trajectory $\tau$ and ultimately determine the quality of the final output $\mathbf{z}_{t_0}$.

# 3. Analysis on Temporal Dynamics in Non-Autoregressive Decoding

In this section, we analyze the unique characteristics of non-autoregressive decoding and characterize proximity bias in 3.1. We further observe the pivotal role of the initial position selection $\mathcal{U}_1$ in steering the subsequent denoising process in 3.2. While we base our primary analysis on LLaDA 8B Instruct (Nie et al., 2025), we demonstrate that these findings generalize to other models (e.g., Dream 7B Instruct (Ye et al., 2025b)) and additional datasets in Appendix C. Experiment details are in Appendix B.1.

## 3.1. Proximity Bias in NAR Decoding

In diffusion-based Large Language Models (dLLMs), generation quality generally improves as the number of decoding timesteps increases, thus allocating more inference-time

---

[1] We employ a linear time schedule as a baseline where the continuous interval is discretized as $t_i = \frac{i}{T}$ for $i = T, T-1, \ldots, 0$.

[2] To simplify the notation for the sequential decision process, we denote the $d$-th decoding step as $d \in \{1, \ldots, T\}$, which corresponds to the transition from $\mathbf{z}_{t_{T-d+1}}$ to $\mathbf{z}_{t_{T-d}}$. Under this convention, $\mathcal{U}_1$ denotes the first position selection made at $t_T$, and the full generation trajectory is defined as $\tau = (\mathbf{z}_{t_T}, \ldots, \mathbf{z}_{t_0})$.

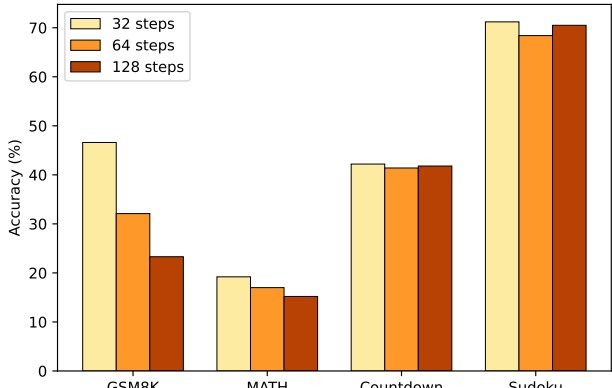

*Figure 2.* Performance in confidence-based non-autoregressive decoding across different diffusion timesteps($T$) when the generation length($L$) is fixed at 256.

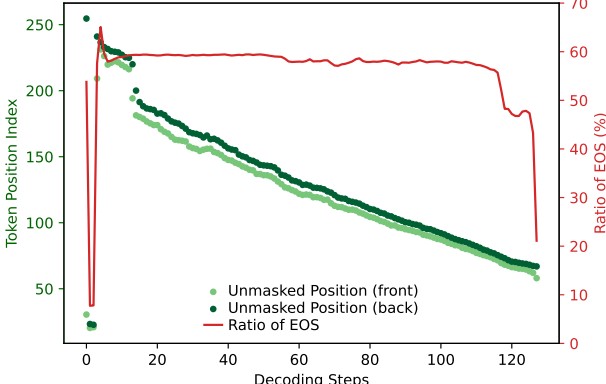

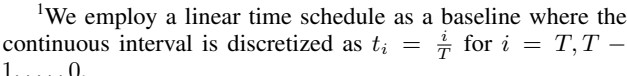

*Figure 3.* Unmasked token position index and average ratio of predicting EOS token across diffusion timesteps (x-axis) when evaluating on GSM8K testset with $T = 128$, $L = 256$. Since two tokens are unmasked simultaneously at each step, we plot both positions, distinguishing between the earlier(front) and later(back) tokens in the sequence. Token position index and ratio of EOS is averaged over all test instances.

computation (Wu et al., 2025a). This behavior is particularly evident in semi-autoregressive (Semi-AR) decoding (Arriola et al., 2025), where each decoding block requires a sufficient number of diffusion steps to achieve stable denoising (Seo et al., 2025), imposing a fundamental constraint in maximizing speed.

In contrast, this monotonic relationship between decoding timesteps and performance breaks down under a non-autoregressive (NAR) regime. Figure 2 illustrates this phenomenon across both mathematical reasoning tasks (GSM8K (Cobbe et al., 2021b), MATH (Lightman et al., 2023)) and planning-intensive tasks (Countdown (Pan et al., 2025), Sudoku (Arel, 2025)), reporting final performance as a function of total decoding timesteps ($T$) with fixed sequence length ($L = 256$). Surprisingly, a smaller number of

steps yields better results, implying that increased inference compute does not necessarily translate to better decoding outcomes in the NAR regime. Motivated by this counterintuitive trend, we analyze the underlying mechanisms using GSM8K as a representative case study.

**End-of-sequence token dominates the initial prediction** Under high uncertainty at the onset of decoding, token-level confidence is largely governed by learned structural priors rather than meaningful semantic comparison (Seo et al., 2025). This causes structural default tokens (e.g., EOS, punctuation, or prompt-specific markers) to be predicted with confidence in early steps, a vulnerability similarly noted in a recent study (Kim et al., 2026). We confirm this behavior by empirically observing that the very last position is preferentially unmasked in the first step. Figure 3 plots the average unmasked token position and average EOS ratio across diffusion steps, revealing that unmasking at the earliest diffusion step occurs at the very last position of the generation window. While Semi-AR decoding circumvents this via imposing block ordering as a stronger structural prior, this phenomenon creates a fundamental conflict with confidence-based heuristics in the NAR regime.

**Spatially adjacent tokens cumulate confidence sequentially** Extending the spatial analysis to the temporal axis, we observe that unmasking exhibits a strong *proximity bias*: once a token is revealed, nearby tokens tend to acquire high confidence, thus being denoised in close temporal proximity. When deterministic decoding is adopted in the NAR regime, once the final position is often unmasked first, the proximity bias causes subsequent denoising to propagate monotonically toward earlier positions. Figure 3 confirms that newly unmasked tokens are strongly concentrated around positions unmasked in the previous step and that unmasking consistently begins at late positions and descends toward the prompt as diffusion progresses. This finding reveals that standard confidence-based NAR decoding collapses into a reverse-autoregressive process.

**Proximity bias propagates end-of-sequence dominance in NAR** Combining earlier findings of premature EOS selection and proximity bias, we find that the generation trajectory becomes strongly anchored by an early end-of-text decision. This aligns with the observation of *EOS overflow* (Kim et al., 2026), where the model leaves insufficient capacity for meaningful content generation. As shown in Figure 3, the average proportion of EOS tokens among the unmasked tokens remains above 50% for most diffusion steps, with valid content tokens emerging only in the final stages. This behavior is reflected in the effective token count: on average, out of 256 positions, 144.6 are occupied by EOS tokens, significantly reducing the model's usable generation window. Crucially, this effect is amplified in

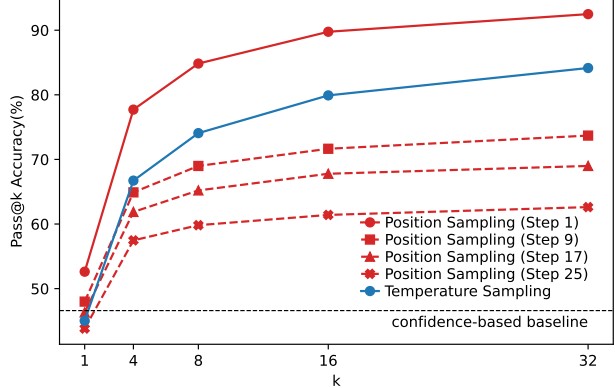

*Figure 4.* Pass@k accuracy on GSM8K with a fixed budget of $T = 32$ across different k(x-axis). The impact of uniform sampling in Position Selection injected at different decoding steps is compared with token-level Temperature Sampling. The solid red line represents randomness applied only at the initial step, while dashed red lines indicate delayed randomness introduced at intermediate steps. The dashed black line denotes the deterministic confidence-based baseline.

a high-compute regime, where only a few tokens are unmasked per step, such that high-probability tokens like EOS dominate the available slots. In contrast, selecting a larger number of tokens per step (i.e., low-compute regime) allows diverse non-EOS candidates to be included, thereby counteracting uncontrolled spatial collapse and reducing the average EOS count to 99. Consequently, this leads to relatively better performance in low-compute setups within the NAR regime. We discuss how this failure mode is mitigated in semi-AR or low-compute NAR regimes in Appendix B.3.

### 3.2. Disproportionate Importance of Initial Unmasking Position Decisions

Building on the analysis of premature end-of-sequence generation and proximity bias, we investigate where and how stochasticity should be introduced in NAR decoding. We focus our investigation on the low-timestep regime($T = 32$, $L = 256$), as it represents the most compute-efficient deployment of dLLMs and exhibits more robust generation in NAR regime.

**Greater impact of randomness in position selection than in token prediction** Due to proximity bias in the NAR regime, we hypothesize that diversity introduced at the earliest step propagates more effectively than stochasticity applied uniformly across steps. To isolate the role of early decisions, we compare two strategies for injecting randomness under identical inference budgets. The first applies temperature-based sampling to token prediction throughout the entire denoising process, where we set temperature as 0.9, following Wang et al. (2025). The second intro-

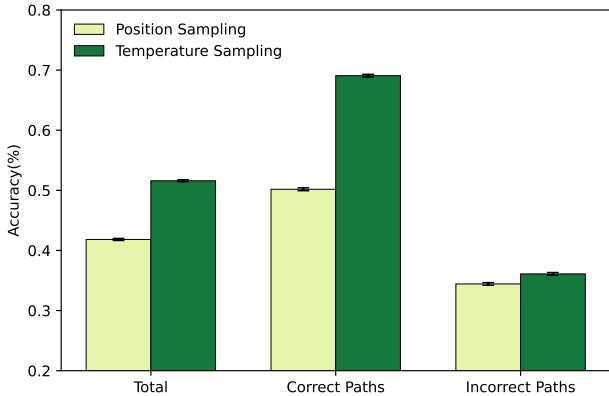

*Figure 5.* Accuracy for randomness introduced via Position Sampling and Temperature Sampling, categorized into Correct Paths, Incorrect Paths, and their union. Error bars represent 95% bootstrapped confidence intervals.

duces randomness in the choice of the denoising position only at the *first* step with uniform sampling, while selecting the token value greedily at all steps. In both cases, all the parameters including timesteps, are held constant.

Figure 4 reports the pass@k performance of each strategy across various $k$ values on GSM8K under a low-compute NAR regime($T = 32$ and $L = 256$). Surprisingly, randomizing only the first denoising position yields a substantially higher pass@k performance than applying temperature sampling across all steps. We further observe that delaying the positional randomization to intermediate steps (dashed red lines) leads to severe performance degradation. These findings highlight that uncertainty in non-autoregressive decoding is highly asymmetric across timesteps. Once an initial denoising decision is made, proximity bias rapidly constrain the subsequent trajectory, limiting the effectiveness of later stochastic perturbations.

**Early Trajectories decisively determine the final generation** A central question is how strongly early denoising decisions constrain the remainder of the generation process. To quantify the decisiveness, we fix initial trajectories and measure the variability introduced by late-stage stochasticity.

For each problem in GSM8K, we construct a two-stage sampling process:

- **Initial Trajectory Anchoring**: We generate 256 distinct trajectories, $\{\tau(\mathcal{U}_1^n)\}_{n=1}^{256}$, by varying the first-step position selection $\mathcal{U}_1$ and applying greedy decoding thereafter. These are categorized into Correct or Incorrect paths based on their final accuracy.
- **Late-stage Stochasticity**: From each category, we sub-

sample up to 16 anchor paths[3]. For each anchor path, we fix the prefix trajectory up to 4 steps ($d = 1, \cdots, 4$), denoted as $\tau_{1:4}$, then generate 8 conditional trajectories $\{\tau^{n,m} \mid \tau_{1:4}^n\}_{m=1}^8$ by introducing stochasticity from $d = 5$ onward (totaling $32 \times 8 = 256$ final samples).

We report accuracy statistics separately for samples originating from correct paths, incorrect paths, and their union. As shown in Figure 5, correct and incorrect trajectories exhibit a substantial performance gap (50.2 vs. 34.4 in random position sampling and 69.1 vs. 36.1 in temperature sampling), with the overall average lying between the two. We also observe that the 95% confidence intervals (error bars) are narrow and non-overlapping, confirming the statistical significance of this separation. Notably, the average accuracy of temperature sampling (51.6) significantly exceeds the baseline temperature sampling performance of 45.0 (Pass@1, blue line in Figure 4).[4] This gain confirms that anchoring the generation, even with a uniformly sampled initial unmasking position independent of the model's greedy confidence, is crucial for stability. In contrast, the average accuracy of random position selection (41.8) proves inferior to the baseline, where position sampling is restricted only to the first step (52.5, Pass@1, red solid line in Figure 4). This degradation implies that position selection is disproportionately critical in the early stages, where continuous randomization disrupts the reasoning structure.

## 4. Initial Trajectory Shaping

Given the pivotal role of the initial position selection in shaping the subsequent denoising process, we propose a lightweight planner designed to steer the model (Wagenmaker et al., 2025) toward a correct denoising path from the very first step. We formally define the problem(4.1) and the methodology(4.2), and sequentially present the experiment details(4.3), results(4.4), and ablation study(4.5).

### 4.1. Problem Definition

The entire trajectory during inference is highly sensitive to early decisions; we denote $\tau(\mathcal{U}_1)$ to represent a trajectory conditioned on the initial position selection. While random sampling or a confidence-based heuristic is typically employed, we instead view position selection as a decision problem that influences the entire future generation.

Let $R(\mathbf{z}_0) \in \{0, 1\}$ denote a task-level reward (e.g., correctness on a reasoning problem). Our goal is to select denoising

---

[3]If fewer than 16 trajectories of a given type are available, all such trajectories are included, resulting in a slightly larger number of incorrect trajectories overall.

[4]The difference between these two is whether initial unmasking token selection $\mathcal{U}_1$ is randomly selected or not and whether tokens were sampled via temperature during initial 4 steps.

positions that maximize expected final performance under fixed subsequent decoding, with specific focus on initial states. Formally, we seek

$$\mathcal{U}^{\star} = \arg \max_{\mathcal{U}} \ \mathbb{E}_{\tau \sim p_{\theta}(\cdot | \mathcal{U}_1)} \big[ R(\mathbf{z}_0) \big]$$

where $\mathcal{U}_1$ is constrained by a fixed budget $|\mathcal{U}_1| = B$, and all denoising steps follow a fixed inference policy in token selection(e.g., greedy or stochastic).

## 4.2. Trajectory Guidance

To mitigate the pathological behaviors identified in Section 3, we propose two strategies for intervening in the generation trajectory focused on initial steps. Our approach aims to guide the model toward semantically consistent paths while preventing premature commitment to structural priors.

**Strategy 1: Early Trajectory Scoring via a Lightweight Planner**   Since evaluating the above objective exactly is intractable, we introduce a *path planner* $\pi_{\phi}(\mathcal{U}_1 \mid h_{\mathcal{S}})$ that predicts optimal denoising positions at the *first* timestep. Crucially, the planner operates on a restricted input: it receives only the final-layer hidden representations $h_{\mathcal{S}}$ corresponding to the candidate sampled positions $\mathcal{S}$, rather than the full sequence context. The planner is trained to assign higher probability to subsets that lead to higher final reward: $\max_{\phi} \ \mathbb{E}_{\mathcal{U}_1 \sim \pi_{\phi}(\cdot | h_{\mathcal{S}})} \big[ R(\mathbf{z}_0) \big]$. Note that the diffusion model parameters $\theta$ are kept fixed, and the planner is designed as a lightweight module with 5M parameters.

During inference, we randomly sample $P$ candidate sets of denoising positions $\{\mathcal{S}^i\}_{i=1}^{P}$ of $\mathcal{U}_1$ and select the one with the highest planner score for the first denoising step. To isolate the effect of the planner and avoid confounding randomness, all subsequent steps are decoded using confidence-based greedy denoising. Importantly, since our method intervenes only at the first step, it is orthogonal to and compatible with other sampling heuristics.

We adopt a progressive denoising schedule in which the number of tokens unmasked at early diffusion steps is deliberately constrained, i.e., $B < \frac{L}{T}$, and it gradually increases over time. Importantly, this design choice is *not* motivated by improvements in confidence-based decoding alone. Instead, the progressive schedule is introduced to reduce early over-commitment and increase the separability of candidate positions at the first diffusion step. We present pseudocode for training and inference in Appendix D.5 and the time schedule details in D.3.

**Strategy 2: Suppressing Premature Termination via EOS Temperature Annealing**   As discussed in Section 3, the inherent bias of assigning high probability to the EOS

token under high uncertainty may lead to a suboptimal trajectory, due to proximity bias. To mitigate this premature commitment, we apply time-dependent temperature annealing specifically to the EOS token. Concretely, we scale the logit of the EOS token by a temperature value $\lambda_d$[5] before the softmax operation. By initializing $\lambda_d$ at a high value and annealing it down to 1 as generation proceeds, we effectively dampen the likelihood of early termination while preserving natural stopping behavior in later stages. Importantly, we employ these scaled logits solely for ranking confidence scores to determine the unmasking position. Once the positions are selected, the token is predicted using the original raw logits via standard greedy argmax.

## 4.3. Experiment Details

We explain the experiment setup and present more detailed aspects in Appendix from D.1 to D.4.

**Implementation Details**   Architecturally, the planner consists of a 2-layer Transformer encoder with learned positional embeddings, followed by a position-wise scoring head. This head predicts a scalar score for each token, which is then averaged to yield the final score. We optimize the planner using a binary cross-entropy loss with trajectory-level correctness labels.

We construct the training dataset via a one-time offline process. For each instance, we sample random positions at the first step and complete the generation using greedy decoding, resulting in a fixed set of trajectories that differ solely in their initial decisions. We fix the sampling budget for both train data generation $(S)$ and inference $(P)$ at 32. Each trajectory is assigned a binary label (correct or incorrect) based on task-specific evaluation metrics applied to the final output, except for Sudoku, where cell accuracy is adopted following (Wang et al., 2025). We set initial $\lambda_T$ as 3 and anneal it down to 1 linearly.

**Experiment Setup**   We utilize LLaDA 8B Instruct (Nie et al., 2025) and Dream 7B Instruct (Ye et al., 2025b) as a backbone diffusion model. We train the planner and evaluate on reasoning and planning tasks: GSM8K (Cobbe et al., 2021a), MATH (Hendrycks et al., 2021), Countdown (Pan et al., 2025), and Sudoku (Arel, 2025). We follow the train-test splits and prompts of prior works (Wang et al., 2025; Zhao et al., 2025a; Tang et al., 2025), with the exception of adding 1-shot examples for Countdown and Sudoku to ensure valid outputs.

**Baselines**   We compare our method against widely used decoding strategies. Specifically, Top-1 Confidence (Chang

---

[5]We highlight again that $d$ denotes the discrete timestep values($d \in \{1, \dots, T\}$).

*Table 1.* Accuracy of basic sampling methods and our strategy on GSM8K (GSM), MATH, Countdown (CTD), and Sudoku (SDK) under a constrained compute budget ($T = 32$). All results are obtained under the non-autoregressive decoding regime. T.S. indicates randomness in Token Selection, while P.S. denotes randomness in Position Selection. **T** represents Temperature sampling, **L** denotes *Learned* position selection, **E** means EOS token logit annealing, and **U** indicates Uniform sampling. The highest score within each column is marked in **bold**, and the second best is underlined. **Both** includes Planner and EOS temperature annealing.

| Sampling Method | Randomness | | | Dataset | | | | Avg |
|---|---|---|---|---|---|---|---|---|
| | T.S. | P.S. | Step | GSM | MATH | CTD | SDK | |
| Top1 Prob | - | - | - | 46.6 | 19.2 | 42.2 | 71.2 | 44.8 |
| + Planner | - | L | 1st | 55.0 | 22.4 | 44.1 | 65.2 | 46.7 |
| + EOS Temp. | - | E | all | 50.9 | 22.4 | **46.1** | 63.6 | 45.7 |
| **+ Both** | - | L&E | all | 56.8 | 22.8 | 43.8 | 67.0 | 47.6 |
| Prob Margin | - | - | - | 47.2 | 19.6 | 41.4 | **71.7** | 45.0 |
| **+ Both** | - | L&E | all | **58.6** | **23.0** | 45.3 | 69.5 | **49.1** |
| Ancestral | - | U | all | 42.1 | 13.2 | 22.7 | 17.8 | 23.9 |
| Temperature | T | - | all | 45.0 | 19.4 | 43.4 | 69.9 | 44.4 |
| Init. Position | - | U | 1st | 52.6 | 17.6 | 30.1 | 48.3 | 37.1 |

et al., 2022) greedily selects unmasking positions based on the highest token probability, while Probability Margin (Kim et al., 2025a) prioritizes positions with the largest gap between the top-1 and top-2 probabilities. As a stochasticity baseline, Ancestral Sampling (Austin et al., 2021) selects positions uniformly at random, while Temperature Sampling selects a token with a temperature value of 0.9 following (Wang et al., 2025). Additionally, we include Random Initial Position, which selects initial positions $\mathcal{U}_1$ randomly but performs greedy decoding thereafter, to serve as a direct control for our planner's contribution.

### 4.4. Experiment Results

In this section, we analyze the impact of randomness and validate the effectiveness of our proposed methods under a constrained compute budget within the NAR regime. While we primarily discuss the results based on LLaDA 8B Instruct in Table 1, Appendix F demonstrates that our approach successfully generalizes to Dream 7B Instruct, confirming its effectiveness across different architectures.

**Superior Performance with Minimal Intervention**   Table 1 presents the performance of various sampling strategies and our proposed methods on four tasks under a constrained timestep ($T = 32$). Our approach, combining the planner and EOS annealing applied to the standard Top-1 Probability baseline, achieves a remarkable average accuracy of 47.6, significantly outperforming the greedy baseline (44.8). Notably, in GSM8K, it delivers a substantial +10.2 performance gain. Importantly, our method demonstrates universality: when applied to the stronger Probability

Margin, performance is further boosted to 49.1. It is worth emphasizing that this gain is achieved with minimal intervention: our planner intervenes in the unmasking position choice only at the very first step, and EOS annealing merely adjusts a single scalar logit at the EOS token. A detailed analysis of latency and computational cost is provided in Appendix G. Notably, in this budget-constrained setup where Semi-AR methods typically struggle, our approach outperforms Semi-AR decoding (average value of 27.0), the details of which are provided in Appendix E.1.

**Efficacy of Learned Planning over Randomness**   We validate the necessity of a *learned* planner by comparing it with uniform random selection at the initial step (Init. Position). While simple random initialization helps in reasoning tasks like GSM8K ($46.6 \rightarrow 52.6$) by introducing diversity to avoid EOS token unmasking, our learned planner identifies favorable initial trajectories more precisely, further pushing performance to 55.0. More critically, the learned planner exhibits robustness across tasks. In structured tasks like Sudoku, where blind randomness causes severe degradation ($71.2 \rightarrow 48.3$), our planner effectively mitigates this drop (65.2), possibly by learning to respect structural priors, demonstrating that it captures task-specific optimal policies rather than acting as a simple noise generator.

**Synergy with EOS Annealing**   While the planner optimizes the starting trajectory, EOS annealing plays a pivotal role in maintaining generation length. As shown in Table 1, although initial random position sampling(Init. Position) can avoid EOS token selection in the first step, it still suffers from severe degradation(Avg 37.1). In contrast, our EOS annealing prevents premature termination during early high-uncertainty stages, while ensuring the generation window is safely terminated in later steps.[6] This validates that targeted guidance, intervening only at the start and on the EOS token, is far more effective than unconstrained exploration in NAR decoding.

**Task-Dependent Sensitivity to Randomness**   Our analysis reveals that for structured problems, the model relies on strong positional priors, and disrupting this order leads to suboptimal output. In Sudoku, specifically, the reasoning steps in the 1-shot example dictate a rigid structure, making the model highly sensitive to any deviation from its preferred decoding path.(Example 2 in Appendix B.1) We provide a deeper analysis of the impact of the task structure and the rigidity of the example in Appendix E.2

---

[6]Empirically, the combined strategy extends the average effective(non-EOS) token count in GSM8K from 157.2 (Top1 Prob greedy baseline) to 188.6.

*Table 2.* Accuracy of baselines and our method on GSM8K, MATH, Countdown, and Sudoku under higher timestep budget($T = 64, 128$). The highest score within each column is marked in **bold**, and the second best is underlined. **Both** includes Planner and EOS temperature.

| Sampling Method | GSM8K | | Math | | Countdown | | Sudoku | | Average | |
|---|---|---|---|---|---|---|---|---|---|---|
| | 64 | 128 | 64 | 128 | 64 | 128 | 64 | 128 | 64 | 128 |
| Top1 Prob | 32.1 | 23.3 | 17.0 | 15.2 | 41.4 | 41.8 | 68.4 | 70.5 | 39.7 | 37.7 |
| Prob Margin | 32.7 | 25.2 | 17.6 | 18.2 | 46.1 | 44.5 | 71.3 | 73.8 | 41.9 | 40.4 |
| Ancestral | 46.5 | 52.3 | 15.4 | 18.2 | 25.4 | 23.8 | 19.1 | 18.5 | 26.6 | 28.2 |
| Temperature | 38.2 | 26.8 | 18.2 | 15.8 | 41.4 | 43.8 | 69.7 | 70.4 | 41.9 | 39.2 |
| Init. Position | 52.2 | 44.9 | 21.2 | 20.6 | 36.7 | 39.5 | 56.4 | 66.9 | 41.6 | 43.0 |
| + Planner | 53.2 | 49.6 | 21.4 | 20.8 | 46.9 | 45.7 | 70.5 | **75.2** | 48.0 | 47.8 |
| + EOS Temp. | 54.5 | 56.6 | **22.4** | **24.0** | **51.2** | **52.0** | **73.9** | 74.2 | 47.4 | 48.1 |
| **Both** | **61.3** | **61.0** | 22.0 | **27.4** | 46.5 | 48.4 | 69.9 | 74.3 | **50.2** | **52.8** |

### 4.5. Ablation Study

**Does the trained planner generalize effectively to larger compute budgets?** We investigate whether our planner, trained on a smaller timestep of $T = 32$, can generalize to inference settings with larger compute budgets ($T = 64, 128$). Even though the planner is applied only at the initial step, its impact is profound. As shown in Table 2, baseline methods exhibit a distinct disparity: while introducing token-level stochasticity with Temperature Sampling (41.9 and 39.2 for $T = 64, 128$, respectively) often maintains or slightly improves performance over greedy decoding(39.7 and 37.7), continuous positional randomness (26.6 and 28.2, Ancestral Sampling) causes severe degradation, particularly in structured tasks. In contrast, our method outperforms all baselines across all tasks and budgets. This result suggests that the planner captures signals about favorable early denoising decisions that generalize across larger diffusion steps, whereas unguided baselines suffer from premature EOS generation at larger $T$, as discussed in Section 3.1.

**How does the candidate pool size ($P$) impact the planner's performance and efficiency?** By comparing performance varying with the candidate size $P$ from 1 to 256, we observe that scaling $P$ beyond 32 incurs additional computational overhead without yielding meaningful performance gains. Thus, we fix $P = 32$ for all experiments as the most cost-effective setting. The result is visualized in Figure 19 and analyzed in detail in Appendix E.3

**Can our approach be applied orthogonally to other baseline heuristics?** Since our method intervenes solely for position selection exclusively in the initial trajectory, it remains orthogonal to other sampling heuristics. We empirically validate this modularity in Appendix E.4, demonstrating that our approach enhances performance on reasoning-intensive tasks.

## 5. Related Work

### 5.1. Diffusion-based Language Models

Early work, such as D3PM (Austin et al., 2021), extended continuous diffusion models (Ho et al., 2020; Song et al., 2021) to discrete state spaces by defining categorical forward noising and reverse denoising processes. Subsequent studies further refined the formulation of discrete diffusion. Campbell et al. (2022) modeled the forward and backward processes over discrete variables as continuous-time Markov chains, enabling principled derivation of training objectives and sampling procedures, while Lou et al. (2023) proposed a score entropy that extends score matching to discrete spaces. Recent studies (Ou et al., 2024; Sahoo et al., 2024; Shi et al., 2024) have simplified the discrete diffusion objective via the absorbing-state property, offering a unified framework aligned with masked generative models. This line of work demonstrates competitive text generation with substantially reduced modeling complexity. Building on these foundations, recent works (Nie et al., 2025; Gong et al., 2025; Ye et al., 2025b) have scaled masked diffusion language models to a billion-parameter scale, achieving performance comparable to autoregressive models.

### 5.2. Efficient Inference for Diffusion-based Large Language Models

To mitigate the trade-off between inference compute and generation quality, recent works have focused on reducing latency through architectural optimizations and dynamic sampling strategies. One line of research leverages Key-Value (KV) caching to lower computational costs (Ma et al., 2025; Liu et al., 2025b; Wu et al., 2025a). However, these methods necessitate a pre-defined unmasking order, typically limiting them to semi-autoregressive regimes to maintain cache validity.

Another line of research adapts the sampling process dynamically based on model's certainty (Yu et al., 2025; Wu et al., 2025b; Ben-Hamu et al., 2025; Li et al., 2025; Kim et al., 2025b; Wei et al., 2025) during generation to harness parallel generation. Kim et al. (2025b) selects tokens to unmask by considering confidence volatility in the token space, while Wei et al. (2025) employs a dual-mode decoding strategy that alternates between slow and fast modes with different unmasking granularities and adaptive local attention windows. Israel et al. (2025) further introduces a small auxiliary language model to select subsets of tokens for parallel generation, while maintaining a strictly autoregressive decoding order. Crucially, these methods predominantly operate under autoregressive dependency to ensure coherence, leaving the potential of fully non-autoregressive dynamics underexplored. In contrast, we focus on the underexplored fully non-autoregressive regime, aiming to unlock the maximum potential for parallel acceleration by eliminat-

ing block-wise constraints entirely.

### 5.3. Sampling Dynamics and Planning in NAR regime

Despite the theoretical appeal of non-autoregressive (NAR) decoding, in-depth analyses of its failure modes in dLLMs remain limited. A notable exception is Seo et al. (2025), who attributes the degradation in NAR generation to the *long decoding-window problem*, where tokens distant from the valid context fail to align with the prompt. To mitigate this, they propose a convolutional filter to enforce local consistency, effectively encouraging smoother confidence accumulation around unmasked regions. While their work primarily addresses spatial incoherence, we extend this analysis to the temporal axis.

Concurrent to our work, Kim et al. (2026) identifies a similar vulnerability in instruction-tuned dLLMs termed *EOS overflow*, where models prematurely terminate generation or degenerate into streams of EOS tokens. To alleviate this, they propose *Rainbow Padding*, which replaces EOS placeholders with distinct padding tokens, inherently requiring fine-tuning of the base dLLM to adapt to the new padding scheme. In contrast, our analysis traces this EOS dominance to the inherent *proximity bias* during the unmasking process. Rather than modifying the base model or introducing new padding representations, we propose a minimal intervention that mitigates this bias by guiding early token selection during unmasking, while leaving the pretrained dLLM unchanged.

Another line of research explores learning-based strategies to guide the unmasking process, employing trained planner models to predict optimal denoising tokens (Peng et al., 2025; Liu et al., 2025a) or corrector models to identify and revise errors (Zhao et al., 2025b; Meshchaninov et al., 2025). While these methods necessitate activating an auxiliary neural network at every timestep, our approach prioritizes minimal intervention, applied solely at the first step based on insights from our spatiotemporal analysis of dLLM behavior. By strictly confining the planner's role to the initial phase, we achieve alignment with the model's generative priors while keeping both training and inference overheads negligible compared to fully guided or corrective baselines.

### 6. Limitation and Discussion

While our analysis and proposed methods demonstrate significant improvements in reasoning and planning tasks, we have not fully explored their efficacy on open-ended text generation, such as creative writing or long-form summarization. Future work should investigate whether the observed proximity bias and the effectiveness of initial trajectory shaping hold in more diverse linguistic contexts, leveraging LLM-as-a-judge frameworks to comprehensively evaluate

generation quality. Further, a systematic exploration of alternative planner designs, such as token-wise selection based on full-sequence embeddings or integrating online trajectory rollouts, could yield further improvements in trajectory optimization.

### 7. Conclusion

In this work, we investigate the under-explored dynamics of fully non-autoregressive decoding through empirical spatiotemporal analysis, revealing that proximity bias causes initial unmasking decisions to disproportionately constrain the final generation trajectory. Leveraging this insight, we propose a strategic framework that employs a lightweight planner and EOS temperature annealing to optimize the unmasking position selection in the initial steps. Our experiments demonstrate that this minimal intervention effectively mitigates structural collapse, yielding significant performance improvements on reasoning-intensive tasks while preserving the efficiency of parallel decoding.

### Impact Statement

This paper presents work whose goal is to advance the field of machine learning. There are many potential societal consequences of our work, none of which we feel must be specifically highlighted here.

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

## A. Preliminaries

We present a brief overview of the Masked Diffusion Language Model (MDLM), adopting the notations in (Sahoo et al., 2024). In MDLM, a clean data is corrupted to a masked state by progressively replacing each token with a special [MASK] token following an absorbing process (Austin et al., 2021). Formally, let $\mathbf{x} \in \{0,1\}^{|\mathcal{V}|}$ denote a one-hot vector representing a single token, and $\mathbf{m}$ denote the [MASK] token vector. During the forward process, $\mathbf{x}$ transitions to $\mathbf{m}$ over a continuous timestep $t \in [0,1]$. $\mathbf{z}_0$ corresponds to the original token $\mathbf{x}$, while $\mathbf{z}_1$ is equal to $\mathbf{m}$. The marginal distribution of $\mathbf{z}_t$ conditioned on $\mathbf{x}$ is defined as:

$$q(\mathbf{z}_t|\mathbf{x}) = \mathrm{Cat}(\mathbf{z}_t; \alpha_t \mathbf{x} + (1 - \alpha_t)\mathbf{m})$$

where the predefined noise schedule $\alpha_t \in [0,1]$ is monotonically decreasing, with $\alpha_0 = 1$ and $\alpha_1 = 0$.

In simplified masked diffusion models (Sahoo et al., 2024; Shi et al., 2024; Ou et al., 2024), due to the absorbing nature of the masking process, the posterior is simplified to:

$$q(\mathbf{z}_s|\mathbf{z}_t, \mathbf{x}) = \begin{cases} \mathrm{Cat}(\mathbf{z}_s; \mathbf{z}_t) & \text{if } \mathbf{z}_t \neq \mathbf{m} \\ \mathrm{Cat}(\mathbf{z}_s; \frac{(1-\alpha_s)\mathbf{m}+(\alpha_s-\alpha_t)\mathbf{x}}{1-\alpha_t}) & \text{if } \mathbf{z}_t = \mathbf{m} \end{cases}$$

A neural network is trained to learn this reverse process by parameterizing the posterior with $p_\theta(\mathbf{z}_s|\mathbf{z}_t) := q(\mathbf{z}_s|\mathbf{z}_t, \hat{\mathbf{x}}_\theta(\mathbf{z}_t, t))$. The learning objective is to minimize the Negative Evidence Lower Bound (NELBO), defined as the weighted integral of the log-likelihood for clean data $\mathbf{x}$:

$$\mathcal{L} = \mathbb{E}_q\left[\int_0^1 \frac{\alpha_t'}{1-\alpha_t} \mathbb{1}[\mathbf{z}_t = \mathbf{m}] \left(\mathbf{x} \cdot \log \hat{\mathbf{x}}_\theta(\mathbf{z}_t, t)\right) \mathrm{d}t\right]$$

where $\alpha_t'$ denotes the time derivative of $\alpha_t$. The expectation is taken over $\mathbf{x} \sim q_0$ and $\mathbf{z}_t \sim q_t(\mathbf{z}_t|\mathbf{x})$. Assuming token-wise independence in the forward process, NELBO can be generalized to a sequence of length $L$ by summing over all tokens:

$$\mathcal{L} = \mathbb{E}_q\left[\int_0^1 \frac{\alpha_t'}{1-\alpha_t} \sum_{l=1}^L \mathbb{1}\left[\mathbf{z}_t^l = \mathbf{m}\right] \left(\mathbf{x}^l \cdot \log \hat{\mathbf{x}}_\theta^l(\mathbf{z}_t, t)\right) \mathrm{d}t\right]$$

## B. Analysis

### B.1. Experiment detail

**Model and Decoding** We conduct the experiments using LLaDA 8B Instruct (Nie et al., 2025) and Dream 7B Instruct (Ye et al., 2025b). Unless otherwise specified, we employ greedy decoding (Top-1 probability). For experiments involving randomness, we use a temperature sampling of 0.9 for token prediction and uniform sampling for position selection strategies.

**Diffusion Configuration**    We adopt a linear noise schedule and evaluate performance across three distinct step budgets: $T \in \{32, 64, 128\}$. The generation sequence length is fixed at 256 tokens for all experiments, as extended lengths did not result in performance improvements for the selected reasoning tasks.

**Prompts and Baselines**    We adopt the basic inference prompts and evaluation code from Wang et al. (2025) with a key modification to the few-shot settings. We found that the standard NAR baseline suffered from severe performance collapse on Sudoku and Countdown under the original settings (0-shot or 3-shot). To ensure valid generation for comparison, we standardized both Sudoku and Countdown to a 1-shot setting. The 1-shot example for Countdown and Sudoku in Example 1 and Example 2, respectively.

## B.2. Dataset

GSM8K (Cobbe et al., 2021b) is a high-quality math word problems with high-quality answer annotation. Each answer has intermediate thought process and the final answer. MATH (Lightman et al., 2023) is composed of challenging competition-level mathematics problems covering a broad spectrum of disciplines, ranging from elementary algebra to calculus and geometry. Countdown (Pan et al., 2025) is an arithmetic reasoning task that requires generating a valid equation using three input integers to derive a specific target value. Sudoku (Arel, 2025) is a logic-based puzzle characterized by strict structural constraints, where the goal is to complete a grid such that every row, column, and subgrid contains unique digits.

## B.3. Inference dynamics in low-compute NAR and Semi-AR

In Section 3.1, we demonstrated that standard non-autoregressive decoding in high-compute regime suffers from a proximity-driven collapse, where the model prematurely anchors to the EOS token at the final position. In this section, we analyze how this dynamic shifts under two alternative setups: the low-timestep regime (which selects more tokens per step) and the semi-autoregressive regime (which enforces structural order).

**Low-Timestep Regime**($T = 32$)    Figure 6 illustrates the unmasking dynamics under a low-timestep budget ($T = 32$). Unlike the high-timestep regime where the model is forced to pick only a few highest confidence tokens, often EOS tokens, the low-timestep setting unmasks a significantly larger volume of tokens at each step. Crucially, as shown in the initial steps of Figure 6, this broader selection window allows valid content tokens adjacent to the prompt to be unmasked alongside the EOS tokens. Once these valid anchors are established near the prompt, the proximity bias

works constructively: subsequent denoising steps expand from these meaningful anchors of generated content rather than propagating solely from the end of the sequence. Consequently, the EOS dominance is naturally diluted; the ratio of EOS tokens peaks at approximately 40% and steadily declines.

**Semi-Autoregressive Regime**($T = 128$)    Figure 7 depicts the dynamics of semi-autoregressive decoding, where the unmasking order is explicitly constrained by a pre-defined schedule, ensuring that tokens are generated in sequential blocks. In this setup, proximity bias is confined within the active block. Even if the model unmasks multiple tokens simultaneously (e.g., 2 tokens per step), the structural constraint forces the generation to proceed strictly from left to right. As a result, the unmasking position moves linearly relative to the decoding steps, effectively mimicking autoregressive behavior. Notably, the ratio of EOS token remains negligible throughout the majority of the generation process and only rises in the final blocks as the sequence naturally concludes. This confirms that while semi-autoregressive constraints effectively mask the proximity bias failure, they do so by reverting to a sequential paradigm, thereby sacrificing the bidirectional flexibility inherent to diffusion models.

## B.4. Detailed Depiction of Proximity Bias

To further scrutinize the spatial dynamics of confidence accumulation, we visualize the evolution of the model's predicted Top-1 probability at each token position across diffusion steps. Figures 8 to 10 present heatmaps where the x-axis represents the diffusion timesteps and the y-axis represents the token position index (0 being the token position right next to the prompt). Across all three regimes, a consistent pattern emerges: confidence propagates continuously from previously unmasked regions to their immediate spatial neighbors, providing a visual confirmation of proximity bias.

In high-budget($T = 128$) NAR setting (Figure 8), the propagation direction is effectively inverted. At the onset of decoding, the region corresponding to the prompt (low y-axis indices) remains low-confidence (dark blue), while high confidence rapidly accumulates at the end of the sequence. As diffusion progresses, the high-confidence region expands downwards from the end of the sequence toward the prompt, visualizing the right-to-left generation anchored by the final EOS tokens propagating backward. While the low-budget($T = 32$) NAR setting (Figure 9) displays a noisier trend with higher number of tokens unmasked each step, the similar trend is observed. On the other hand, in the semi-autoregressive setting (Figure 10), the confidence propagation exhibits a distinct, sharp diagonal trajectory moving from lower to higher token indices aligning perfectly with the pre-defined block-wise schedule.

---

**Prompt 1: 1-shot Example for Countdown**

```
Question:
Numbers: [37, 89, 41]
Target: 11
Answer:
<reasoning>
Let's break down the steps:

1. Start with the largest number, 89, and try to use it in the expression.
2. Use the subtraction operation to get the target number 11.

Let's try:
- 89 - 37 = 52
- 52 - 41 = 11

So, the expression is 89 - 37 - 41 = 11.

This expression uses each number exactly once and evaluates to the target
number 11.
</reasoning>
<answer>
\boxed{89 - 37 - 41}</answer>
```

---

*Table 3.* Performance of Dream 7B Instruct in non-autoregressive decoding with confidence criteria across different diffusion timesteps($T$) when the generation length is fixed at 256. Effective token counts reported in parentheses.

| $T$ | GSM8K | MATH | Countdown | Sudoku |
|---|---|---|---|---|
| 32 | 39.9 (101.8) | 17.2 (78.6) | 50.0 (121.8) | 7.5 (156.2) |
| 64 | 45.5 (78.9) | 17.0 (35.9) | 50.8 (121.2) | 8.6 (147.5) |
| 128 | 33.9 (47.8) | 17.2 (22.9) | 49.6 (120.4) | 0.0 (5.8) |

## C. Generalization of Temporal Dynamics Analysis

In this section, we extend the analysis from Section 3 to verify that the observed spatiotemporal dynamics generalize across different model architectures and datasets. Unless otherwise specified, the experimental setups remain identical to those in the main text.

### C.1. Results on a Different Architecture: Dream 7B

**More steps do not translate to better performance.** Table 3 presents the performance of Dream 7B Instruct (Ye et al., 2025b) across varying timesteps, with effective (non-EOS) token counts reported in parentheses. We observe that increasing the timestep budget invariably reduces the number of effective tokens, ultimately leading to severe performance degradation at high-compute budget (e.g., $T = 128$).

**Proximity bias and EOS dominance** Figure 11 and 12 illustrate the unmasking dynamics of Dream 7B Instruct when $T = 128$ and $T = 32$, respectively. Dream model also exhibits premature EOS token selection at the onset of decoding, which cascades via proximity bias and severely limits the capacity for valid content generation.

**Position selection randomness outweighs token prediction randomness.** Figure 13 reports the pass@k performance of Dream 7B Instruct on GSM8K in the low-step NAR regime ($T = 32$ and $L = 256$). We compare the effects of injecting randomness via token-level temperature sampling at all steps versus randomizing the denoising position exclusively at the first step. The results on Dream 7B Instruct demonstrate that applying randomness to the initial position steers the generation trajectory significantly better than token-level sampling at all steps.

**Trajectory shaped early decisively determines the final generation.** Figure 14 presents the results of the two-stage trajectory anchoring experiment applied to Dream 7B Instruct. For computational efficiency, we use a scaled-down setup, generating 32 initial trajectories and sub-sampling up to 4 anchor paths per category. Consistent with our observations in Section 3.2, we observe a stark performance gap between late-stage generations anchored to initially correct versus incorrect trajectories. This confirms that early denoising decisions decisively shape the final output regardless of

---

**Prompt 2: 1-shot Example for Sudoku**

```
Question:
Solve the following Sudoku puzzle: 3014002020004130
Answer:
<reasoning>
Interpret puzzle as 4 rows of 4:
R1: 3 0 1 4
R2: 0 0 2 0
R3: 2 0 0 0
R4: 4 1 3 0

Fill easy singles:
R1 missing 2 → R1C2=2.
R4 missing 2 → R4C4=2.
Box D (R3-4,C3-4) then needs {1,4}; column4 can only accept 1 → R3C4=1, R3C3=4.
R3 now missing 3 → R3C2=3.
Column1 missing 1 → R2C1=1.
Column2 missing 4 → R2C2=4.
Last cell R2C4=3.

Final grid:
R1: 3 2 1 4
R2: 1 4 2 3
R3: 2 3 4 1
R4: 4 1 3 2
</reasoning>
<answer>
3214142323414132
</answer>
```

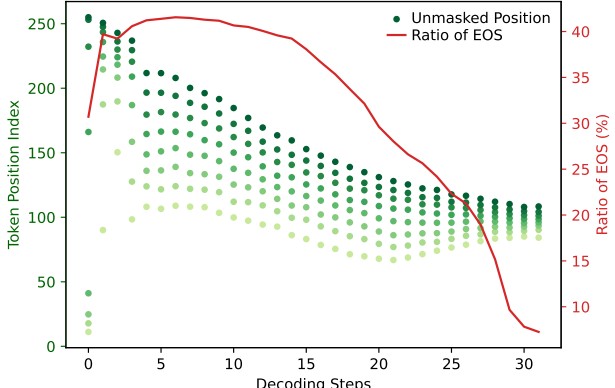
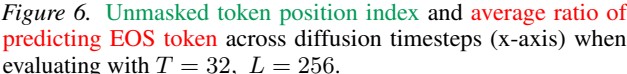

*Figure 6.* Unmasked token position index and average ratio of predicting EOS token across diffusion timesteps (x-axis) when evaluating with $T = 32$, $L = 256$.

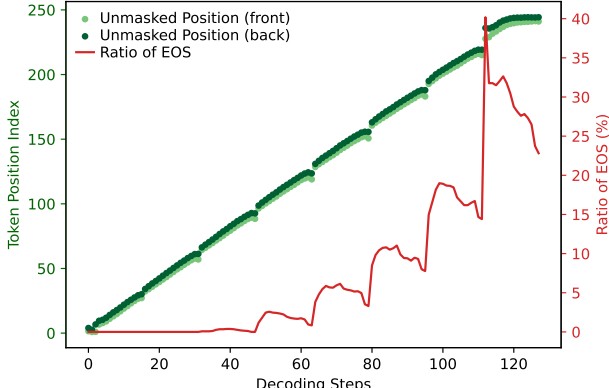

*Figure 7.* Unmasked token position index and average ratio of predicting EOS token across diffusion timesteps (x-axis) with Semi-AR decoding.

late-stage stochasticity across different model architectures.

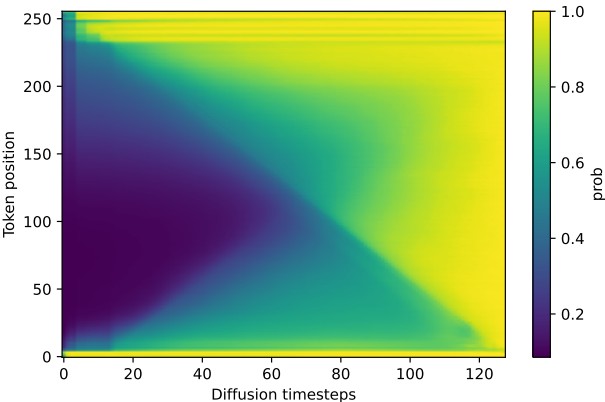

*Figure 8.* Top 1 probability predicted at each diffusion timestep(x-axis) for each token position(y-axis) in Non-Autoregressive decoding under lenient budget($T = 128$).

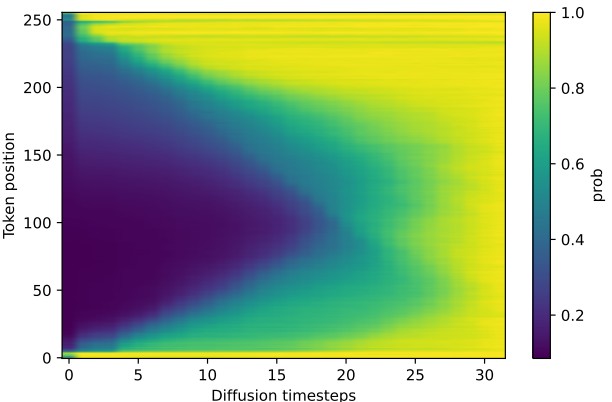

*Figure 9.* Top 1 probability predicted at each diffusion timestep(x-axis) for each token position(y-axis) in Non-Autoregressive decoding under tight budget($T = 32$).

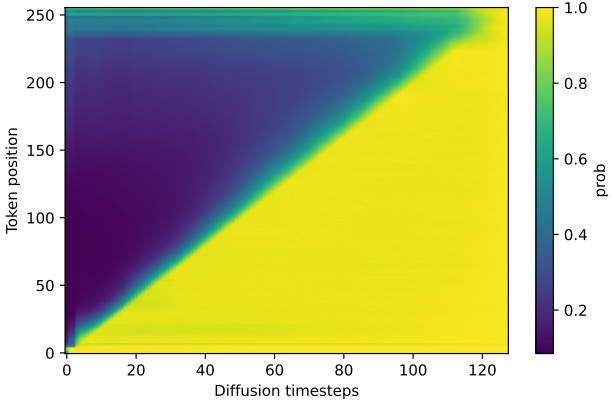

*Figure 10.* Top 1 probability predicted at each diffusion timestep(x-axis) for each token position(y-axis) in Semi-Autoregressive decoding under lenient budget($T = 128$).

## C.2. Results on Additional Datasets: MATH, Countdown, and Sudoku

We evaluate LLaDA 8B Instruct on the MATH, Countdown, and Sudoku datasets to confirm that the observed dynamics

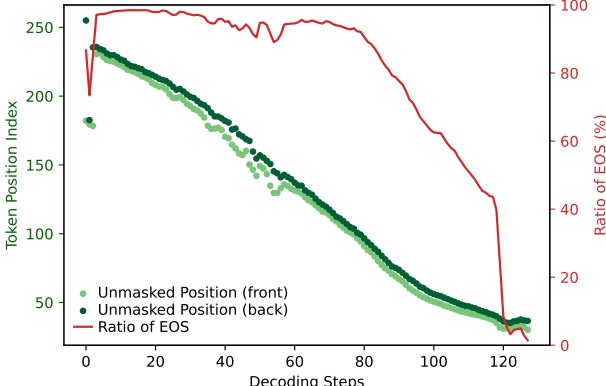

*Figure 11.* Unmasked token position index and average ratio of predicting EOS token across diffusion timesteps (x-axis) when evaluating **Dream 7B Instruct** on GSM8K testset with $T = 128$, $L = 256$.

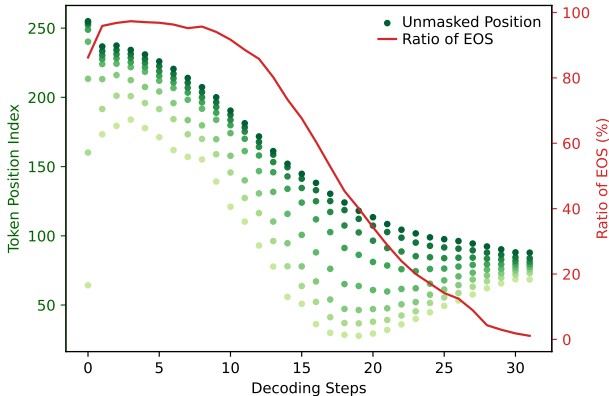

*Figure 12.* Unmasked token position index and average ratio of predicting EOS token across diffusion timesteps (x-axis) when evaluating **Dream 7B Instruct** on GSM8K testset with $T = 32$, $L = 256$.

are not strictly tied to GSM8K.

**Proximity bias and EOS dominance.** Figure 15 illustrates the unmasking dynamics when evaluating on MATH at a high compute budget($T = 128$). In MATH, we observe the same EOS dominance and reverse-autoregressive propagation as seen in GSM8K. In Countdown(Figure 16) and Sudoku(Figure 17), however, the premature EOS tendency is significantly weaker. This strongly aligns with our analysis in Section 4.4 and Appendix E.2 that the strict 1-shot example acts as strong structural priors, effectively mitigating the early EOS collapse.

**Importance of initial unmasking decisions.** Table 4 reports the pass@k performance across the datasets, and Figure 18 replicates the trajectory anchoring experiment specif-

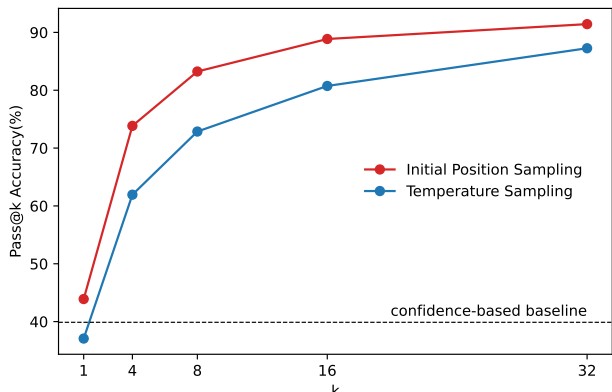

*Figure 13.* Pass@k accuracy of **Dream 7B Instruct** on GSM8K with a fixed budget of $T = 32$ across different k(x-axis). The impact of uniform sampling in Position Selection injected at different decoding steps is compared with token-level Temperature Sampling.

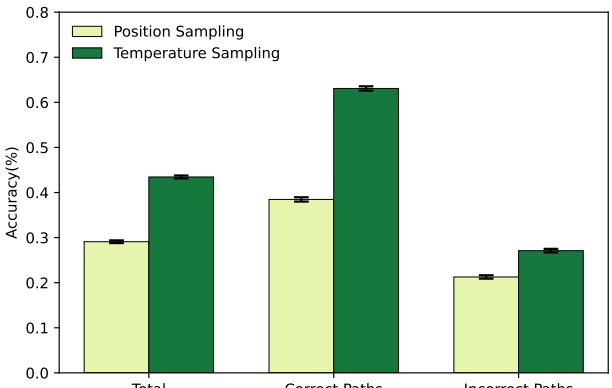

*Figure 14.* Accuracy of **Dream 7B Instruct** for randomness introduced via Position Sampling and Temperature Sampling, categorized into Correct Paths, Incorrect Paths, and their union. Error bars represent 95% bootstrapped confidence intervals.

ically for the MATH dataset.[7] Consistent with our findings in GSM8K(Section 3.2) and Dream 7B(Appendix C.1), the results confirm that randomizing the initial position is far more effective than continuous token-level sampling. Moreover, early denoising decisions rigidly constrain the final generation, demonstrating that this disproportionate importance of initial steps is a fundamental characteristic of the confidence-based NAR decoding process, regardless of the target task.

---

[7]While the procedure remains identical, we use a smaller sample size (32 initial trajectories and up to 4 anchor paths).

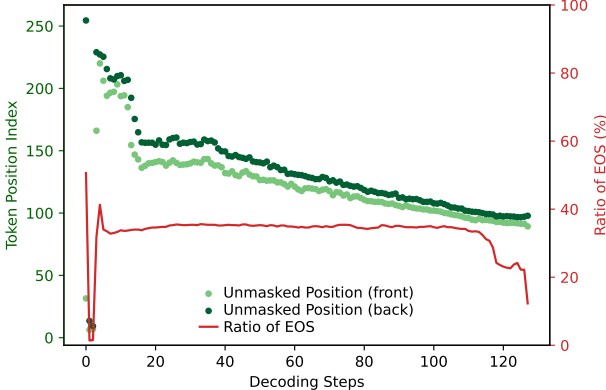

*Figure 15.* Unmasked token position index and average ratio of predicting EOS token across diffusion timesteps (x-axis) when evaluating LLaDA 8B Instruct on **MATH** with $T = 128$, $L = 256$.

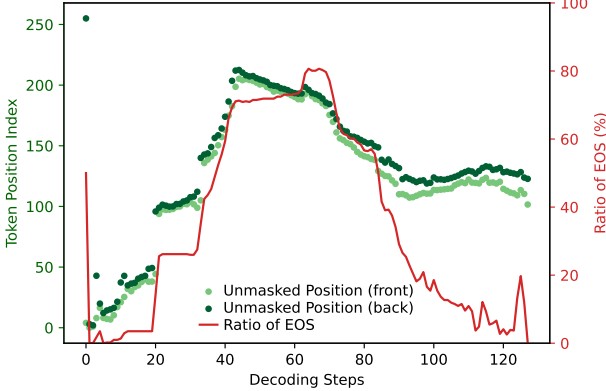

*Figure 16.* Unmasked token position index and average ratio of predicting EOS token across diffusion timesteps (x-axis) when evaluating LLaDA 8B Instruct on **Countdown** with $T = 128$, $L = 256$.

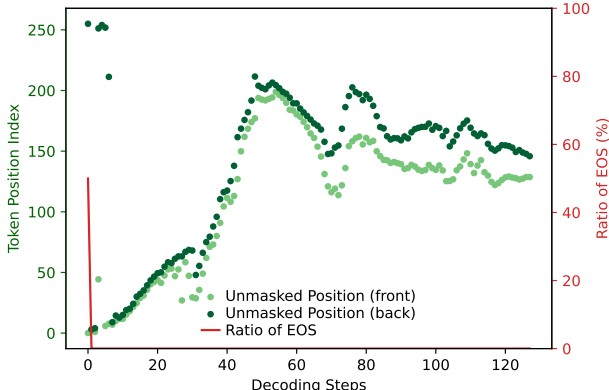

*Figure 17.* Unmasked token position index and average ratio of predicting EOS token across diffusion timesteps (x-axis) when evaluating LLaDA 8B Instruct on **Sudoku** with $T = 128$, $L = 256$.

*Table 4.* Comparison of Pass@k accuracy between uniform Position Selection (P.S.) injected solely at the first step and continuous token-level Temperature Sampling (T.S.).

|  |  | pass@1 | pass@4 | pass@8 | pass@16 | pass@32 |
|---|---|---|---|---|---|---|
| GSM8K | P.S. | 52.6 | 77.7 | 84.8 | 89.8 | 92.5 |
|  | T.S. | 45.0 | 66.7 | 74.1 | 79.9 | 84.2 |
|  | diff | 7.6 | 11.0 | 10.8 | 9.9 | 8.3 |
| MATH | P.S. | 17.6 | 33.4 | 41.0 | 48.6 | 55.2 |
|  | T.S. | 19.4 | 28.4 | 33.2 | 39.2 | 43.4 |
|  | diff | -1.8 | 5.0 | 7.8 | 9.4 | 11.8 |
| Countdown | P.S. | 30.1 | 55.1 | 64.8 | 71.9 | 76.2 |
|  | T.S. | 43.4 | 55.1 | 57.8 | 62.1 | 65.2 |
|  | diff | -13.3 | 0.0 | 7.0 | 9.8 | 10.9 |
| Sudoku | P.S. | 48.3 | 75.3 | 83.7 | 88.8 | 93.2 |
|  | T.S. | 69.9 | 84.2 | 86.6 | 89.7 | 91.6 |
|  | diff | -21.6 | -8.9 | -2.9 | -0.9 | 1.7 |

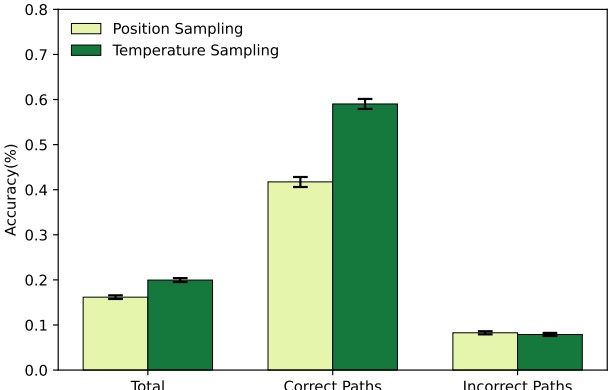

*Figure 18.* Accuracy of LLaDA 7B Instruct on **MATH** for randomness introduced via Position Sampling and Temperature Sampling, categorized into Correct Paths, Incorrect Paths, and their union. Error bars represent 95% bootstrapped confidence intervals.

# D. Experiment Setup of Initial Trajectory Shaping

## D.1. Planner training details

**Architecture Design** We design the planner as an extremely lightweight scoring module with approximately 5M parameters to ensure that it introduces negligible latency during inference. The architecture consists of a 2-layer Transformer Encoder with an input dimension of $d_{model} = 128$. The processing pipeline is as follows:

1. **Input Projection:** The hidden states from the diffusion backbone ($D = 4096$) are first projected down to the planner's dimension ($d_{model} = 128$).

2. **Lightweight Positional Embedding:** Positional embeddings with a low dimension ($d_{pos} = 16$) are projected to $d_{model} = 128$ to be added to the input fea-

tures then input to the transformer layer with ReLU activation in between.

3. **Scoring Head:** The transformer outputs for each token are projected to a scalar value. Final score for the sampled embeddings is obtained as an average of these values.

**Training Configuration** The planner is trained using the Binary Cross Entropy loss. We employ the AdamW optimizer with a fixed learning rate of 1e-4 and a batch size of 256. To prevent overfitting to the training samples, we apply a dropout rate of 0.3 and limit training to a maximum of 5 epochs. The maximum positional embedding length is fixed at 256, aligning with our generation settings. We determined the optimal architecture configurations and training hyperparameters through a grid search, where the final selection was based on the the reranking accuracy on the held-out validation set.

## D.2. Dataset construction details

We follow the inference prompts and evaluation code from Wang et al. (2025) as detailed in B.1. We generate training data via an offline sampling process. For each prompt in the training set, we execute the first diffusion step by randomly sampling the unmasking positions. Crucially, to evaluate the true impact of this initial choice, all subsequent decoding steps are performed using deterministic greedy decoding. We assign binary labels based on the accuracy of the final generated output, with only exception for Sudoku, where soft labels of cell accuracy is employed following Wang et al. (2025). We split the training and validation sets based on unique prompts, ensuring that samples derived from the same prompt do not appear in both splits.

## D.3. Time schedule details

Instead of relying on a standard linear schedule, we adopt a hybrid allocation strategy. To prevent the model from operating on an excessively uncertain context at the onset of decoding, we enforce a strict constraint that at least $w$ tokens are unmasked at every step. After reserving the guaranteed tokens for all $T$ steps, the remaining tokens $N_{resid} = L - (w \times T)$ are distributed according to a power-law schedule. Specifically, the residual tokens are allocated proportional to $t^v$, where $v$ controls the acceleration of denoising. We set $w = 3, v = 1$ for progressive schedule.

## D.4. EOS Temperature Annealing details

To counteract the dominant structural default of premature EOS prediction, particularly prevalent in the high-uncertainty early stages, we introduce a targeted EOS logit annealing strategy. This method dynamically scales the

**Algorithm 1** Planner Training via Offline Trajectory Sampling and Scoring

1: **Input:** Training prompts $\mathcal{D}$, Diffusion backbone $p_\theta$ (frozen), Number of training samples per prompt $S$, Epochs $E$
2: **Output:** Optimized Planner parameters $\phi$
3: *// Phase 1: Offline Data Construction*
4: Initialize dataset buffer $\mathcal{B} \leftarrow \emptyset$
5: **for** each prompt $\mathbf{x} \in \mathcal{D}$ **do**
6:    **for** $s = 1$ **to** $S$ **do**
7:       Sample initial positions $\mathcal{S}^s$ uniformly
8:       Generate: $\hat{\mathbf{z}}_0 \leftarrow \text{GreedyDecode}(p_\theta, \mathbf{x}, \mathcal{S}^s)$
9:       Compute label: $y^s \leftarrow \text{Reward}(\hat{\mathbf{z}}_0)$ *// 1 if correct, else 0*
10:      Add $(\mathbf{x}, \mathcal{S}^s, y^s)$ to $\mathcal{B}$
11:    **end for**
12: **end for**
13: *// Phase 2: Planner Optimization*
14: Initialize planner $\pi_\phi$
15: **for** epoch $e = 1$ **to** $E$ **do**
16:    **for** each batch $\{(\mathbf{x}_i, \mathcal{S}_i, y_i)\}$ in $\mathcal{B}$ **do**
17:       Compute backbone features: $H_i \leftarrow p_\theta(\mathbf{x}_i)$ *// No Gradient*
18:       Extract features at positions: $h_i \leftarrow H_i[\mathcal{S}_i]$
19:       Predict score: $\hat{y}_i \leftarrow \pi_\phi(h_i)$
20:       Update $\phi \leftarrow \text{Optimizer}(\phi, \nabla_\phi \mathcal{L})$
21:    **end for**
22: **end for**
23: **Return** $\phi$

**Algorithm 2** Inference with Planner-Guided Initial Trajectory

1: **Input:** Masked input $\mathbf{z}_T$, Diffusion model $p_\theta$, Planner $\pi_\phi$, Number of candidates $P$, Diffusion timestep $T$
2: **Output:** Generated sequence $\mathbf{z}_0$
3: *// Phase 1: Planner-Guided Initialization ($d = 1$)*
4: Compute backbone features: $H \leftarrow \text{Encoder}_\theta(\mathbf{z}_T)$
5: Sample $P$ candidate position sets($\mathcal{S}$) uniformly at random: $\{\mathcal{S}^i\}_{i=1}^P$
6: Initialize scores list $V$
7: **for** $i = 1$ **to** $P$ **do**
8:    Extract features: $h^i \leftarrow H[\mathcal{S}^i]$
9:    Predict score: $v^i \leftarrow \pi_\phi(h^i)$
10:   Append $v^i$ to $V$
11: **end for**
12: Select optimal positions: $\mathcal{U}_1^* \leftarrow \mathcal{S}^{(\text{argmax } V)}$
13: Unmask tokens at $\mathcal{U}_1^*$ and obtain $\mathbf{z}_{T-1}$ via $p_\theta$
14: *// Phase 2: Standard Decoding ($d = 2 \ldots T$)*
15: **for** $d = 2$ **to** $T$ **do**
16:    Select next positions $\mathcal{U}_d$ via pre-defined position selection strategy
17:    Unmask tokens at $\mathcal{U}_d$ using $p_\theta$ to update $\mathbf{z}_{T-d}$
18: **end for**
19: **Return** $\mathbf{z}_0$

*Table 5.* Accuracy on GSM8K(GSM), MATH, Countdown (CTD), and Sudoku (SDK) under a constrained compute budget ($T = 32$) in Semi-autoregressive and non-autoregressive regime. The highest score within each column is marked in **bold**.

| Sampling Regime | Time Schedule | Dataset | | | | Avg |
|---|---|---|---|---|---|---|
| | | GSM | MATH | CTD | SDK | |
| SemiAR | Linear | 46.5 | 19.8 | 25.4 | 16.2 | 27.0 |
| NAR | Linear | 46.6 | 19.2 | 42.2 | **71.2** | 44.8 |
| NAR | Progressive | 45.0 | 16.4 | 39.5 | 67.4 | 42.1 |
| NAR | **+ Ours** | **56.8** | **22.8** | **43.8** | 67.0 | **47.6** |

logit value of the EOS token to suppress premature termination without altering the semantic distribution of other tokens. We define a time-dependent scaling factor, $\lambda_d$, which follows a linear decay schedule from an initial strong suppression factor to a neutral state of 1. Specifically, we set $\lambda_t = 3 - \frac{2d}{T}$.

### D.5. Algorithmic Details for the Planner

We provide the complete pseudocode for the planner. Please refer to Algorithm 1 for the offline planner training pipeline and Algorithm 2 for the planner-guided inference procedure.

## E. Additional Results of Initial Trajectory Shaping

### E.1. Comparison with Semi-Autoregressive Decoding

We further benchmark our approach against semi-autoregressive decoding strategies under a strictly constrained compute budget ($T = 32$), a regime critical for low-latency applications. As presented in Table 5, Semi-AR methods suffer from a severe performance collapse when

the diffusion step count is reduced especially in planning tasks, yielding an average accuracy of 27.0. In contrast, our method, operating within the fully non-autoregressive regime, maintains robustness, achieving a significantly higher average accuracy of 47.6. This result demonstrates that our planner-guided intervention effectively unlocks the potential of parallel decoding.

### E.2. Task-dependent effect of randomness in position selection and token prediction

Although our approach significantly improves performance over all the baselines, Sudoku is an exception. We analyze that the impact of randomness varies significantly by task structure. Across all tasks, introducing positional randomness at every step yields the lowest performance (average 23.9), marking a significant drop from the greedy Top-1

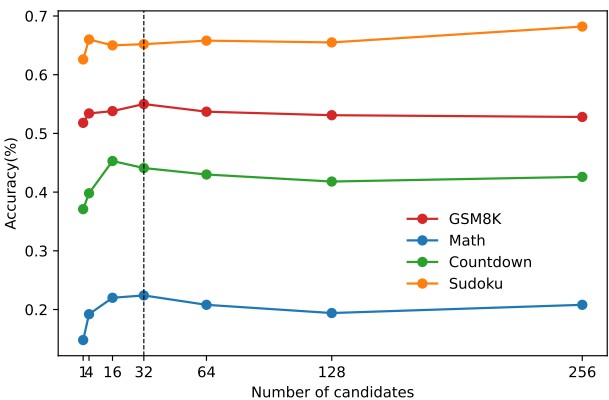

*Figure 19.* Performance on each task across different numbers of candidates $P$.

*Table 6.* Accuracy of basic sampling methods with and without our learned planner on GSM8K, MATH, Countdown, and Sudoku under a constrained compute budget ($T = 32$). All results are obtained under the non-autoregressive decoding regime. The highest score within each column is marked in **bold**, and the second best is underlined. **+Ours** indicates the integration of our learned planner and EOS temperature.

| Sampling Method | Dataset | | | | Avg |
|---|---|---|---|---|---|
| | GSM8K | MATH | Countdown | Sudoku | |
| Top1 Prob | 46.6 | 19.2 | 42.2 | 71.2 | 44.8 |
| **+ Ours** | 56.8 | 22.8 | 43.8 | 67.0 | 47.6 |
| Margin | 47.2 | 19.6 | 41.4 | **71.7** | 45.0 |
| **+ Ours** | **58.6** | **23.0** | **45.3** | 69.5 | **49.1** |
| Ancestral | 42.1 | 13.2 | 22.7 | 17.8 | 23.9 |
| **+ Ours** | 45.9 | 16.8 | 23.4 | 20.3 | 26.6 |
| Temperature | 45.0 | 19.4 | 43.4 | 69.9 | 44.4 |
| **+ Ours** | 55.9 | 22.6 | 31.6 | 52.3 | 40.6 |

Confidence baseline (44.8). This decline is most severe in structured planning tasks like Countdown and Sudoku, even when the perturbation is applied only at the first step (Init. Position, 37.1), while global stochasticity in token selection with temperature sampling results in only a mild drop (44.4). We hypothesize that for structured problems, the model relies on strong positional priors, and disrupting this order leads to suboptimal output. In Sudoku, specifically, the reasoning steps in the 1-shot example dictate a rigid structure, making the model highly sensitive to any deviation from its preferred decoding path.(See Example 2 in Appendix B.1) On the other hand, we observe nuanced behaviors in reasoning tasks. In GSM8K, performance improves only when positional randomness is restricted to the first step, whereas global stochasticity in position (Ancestral) or token (Temperature) leads to degradation. This indicates that early exploration helps find better reasoning trajectories, but the subsequent logical path must be exploited greedily. In contrast, MATH exhibits sensitivity to positional changes while temperature sampling provides a marginal gain.

### E.3. Ablation on Candidate Size $P$

Figure 19 illustrates the performance varying with the candidate size $P$. For GSM8K and MATH, accuracy improves notably as $P$ increases, reaching an optimal plateau at $P = 32$. In contrast, Countdown saturates at a smaller $P$, and Sudoku exhibits a flat trend, reflecting the varying reliance on trajectory planning across tasks. Crucially, scaling $P$ beyond 32 does not yield meaningful performance gains even with additional computational overhead. Thus, we fix $P = 32$ for all experiments as the most cost-effective setting.

### E.4. Compatibility with Existing Sampling Heuristics

Our proposed approach intervenes exclusively in position selection especially in the early stages. Consequently, it remains operationally orthogonal to the specific heuristics employed for token value selection or intermediate unmasking schedules. This modularity allows our method to be seamlessly integrated with various existing sampling strategies. Table 6 summarizes the performance of applying our method, combining the planner and EOS temperature annealing, to the baselines under a constrained budget ($T = 32$).

On mathematical reasoning benchmarks (GSM8K and MATH), our approach yields robust performance improvements across all tested sampling strategies. For Countdown, we observe improvements when our method is paired with greedy token selection, whereas it leads to degradation with temperature sampling. In the case of Sudoku, our intervention generally leads to performance degradation. As discussed in the Section 4.4, the model's intrinsic confidence is already near-optimal based on the highly structured 1-shot example. In such scenarios, external interventions tend to disrupt the strong inductive bias required for the task. Overall, these results highlight that our method can be introduced flexibly and it is particularly effective in open-ended reasoning scenarios where the model's intrinsic confidence may be misplaced due to proximity bias.

## F. Generalization of Initial Trajectory Shaping

To verify the generalizability of our proposed method, we apply our learned planner and EOS temperature to Dream 7B Instruct (Ye et al., 2025b). As shown in Table 7, our approach consistently improves the performance of basic sampling strategies on both GSM8K and MATH under constrained budgets ($T = 32$), demonstrating its effectiveness across different dLLMs. Specifically, consistent with our findings in Section 4.4, the proposed method not only significantly boosts the performance of deterministic baselines (Top-1 Prob and Prob Margin), but also robustly outper-

*Table 7.* Accuracy of basic sampling methods and our learned planner on GSM8K and MATH using **Dream 7B Instruct** under a constrained compute budget ($T = 32$). The highest score within each column is marked in **bold**, and the second best is underlined. **+Ours** indicates the integration of our learned planner and EOS temperature.

| Sampling Method | Dataset | | Avg |
|---|---|---|---|
| | GSM8K | MATH | |
| Top1 Prob | 39.9 | 17.2 | 28.5 |
| + Ours | 52.4 | **21.4** | 36.9 |
| Prob Margin | 40.9 | 16.2 | 28.5 |
| + Ours | **54.7** | 19.6 | **37.1** |
| Ancestral | 28.7 | 12.8 | 20.7 |
| Temperature | 36.1 | 15.0 | 25.5 |
| Initial Position | 43.9 | 15.4 | 29.6 |

forms naive initial random sampling. This confirms that the efficacy of learned planning over simple randomness is a model-agnostic characteristic.

## G. Computational Overhead Analysis

In this section, we provide a detailed analysis of the computational overhead introduced by our proposed method, demonstrating its efficiency in both inference and training phases.

**Inference Latency**    To substantiate our claim of negligible inference overhead, we measured the average end-to-end wall-clock latency of LLaDA 8B Instruct on GSM8K testset($1\times$ A100 GPU, Batch Size=1, $L = 256$, $T = 32$). Our method's average latency (2.54s) is virtually identical to the Top-1 Prob baseline (2.65s), with the minor difference falling well within normal hardware variance. This imperceptible overhead stems from two architectural design choices:

1. $\mathcal{O}(1)$ **Intervention:** The 5M-parameter planner, accounting for less than 0.1% of the 8B backbone, is executed only once at the first denoising step. All subsequent steps revert to the standard Top-1 decoding.
2. **Zero-Cost Annealing:** The EOS temperature annealing requires only a simple scalar multiplication, adding no measurable delay.

**Training Compute Overhead**    Our method is exceptionally lightweight, especially when contrasted with the massive memory and compute requirements of standard model alignment techniques in RLVR or standard RLHF paradigms (Ouyang et al., 2022; Shao et al., 2024). The computational profile consists of two highly asymmetric phases:

1. **Offline Trajectory Generation (Inference-Only):** Sampling training data requires an upfront compute investment but is strictly a forward-pass operation on a frozen backbone. Without the need for gradient computation or optimizer states for the large language model, the peak memory footprint is drastically reduced. Moreover, this is a one-time, highly parallelizable offline process.
2. **Planner Optimization (Lightweight Training):** Since gradients are strictly confined to 5M-parameter planner, the training converges rapidly—taking approximately 5 minutes on a single A100 GPU—with a negligible memory footprint.

By isolating the 8B model entirely to inference and restricting backpropagation exclusively to the 5M planner module, our approach completely bypasses the need for multi-GPU clusters and the extensive compute times typical of standard alignment methods.

