# OpenReview forum: "Early Decisions Matter: Proximity Bias and Initial Trajectory Shaping in Non-Autoregressive Diffusion Language Models"
_ICML.cc/2026/Conference — ICML 2026 regular_

### Official Review · Reviewer_GnZi · 2026-03-08

**Soundness:** 2
**Presentation:** 3
**Significance:** 2
**Originality:** 3
**Overall Recommendation:** 4
**Confidence:** 4

**Summary:**

This paper investigates failure modes of fully non-autoregressive (NAR) decoding in Masked Diffusion Language Models. The authors identify proximity bias, a phenomenon where confidence-based NAR decoding causes spatially adjacent tokens to accumulate confidence sequentially. Combined with the model's tendency to preferentially predict high-probability structural tokens (e.g., EOS) under initial uncertainty, this results in a reverse-autoregressive unmasking pattern that propagates from the sequence end back to the front, severely truncating the effective generation window. Building on the observation that the first denoising step disproportionately determines the final output (temporal asymmetry of importance), the authors propose two lightweight interventions: (1) a 5M-parameter Planner that scores candidate unmasking positions at the first step, and (2) EOS temperature annealing that suppresses premature sequence termination. The proposed framework is orthogonal to the existing training and sampling recipes of dLLMs and hence can be applied in addition to existing pipelines. The authors demonstrated the performance of their pipeline on extensive benchmarks including GSM8K, MATH, Countdown, and Sudoku.

**Compliance With Llm Reviewing Policy:**

Affirmed.

**Final Justification:**

All concerns addressed. I mentioned in the original review that this paper is more reactive than fundamental but such a lightweight module that actually makes meaningful improvements to model training is still novel and interesting. Therefore, I recommend a weak accept.

**Key Questions For Authors:**

Please see above section.

**Limitations:**

Yes

**Strengths And Weaknesses:**

The paper is well written. The characterization of proximity bias is thorough and well-visualized. Section 3 provides insightful empirical analysis into the existing problems that the authors wish to solve.

The experimental design is clean and effective to demonstrate the existing problems.

The proposed method is simple and useful. Although to some extent I find it more reactive than fundamental (meaning position matters → train a model to pick positions; EOS dominates → scale down EOS logit), still since it's lightweight and useful, that's still original and carries some significance.

Some concerns are listed as follows:
1. LLaDA 8B is the only model investigated and I wonder if this phenomenon is indeed generalizable.
2. The planner is model-specific and task-specific and hence must be retrained for every model and every task and it only works for verifiable tasks because we need the binary reward (for example, we cannot apply it to open ended generation in the current setup). I wonder if the training of the planner model causes non-negligible compute overhead. In fact, the authors claim it's lightweight but there is no experiment supporting it.
3. In section 3.2, paragraph Trajectory shaped early decisively determines the final generation, only 32 trajectories are sampled, which seem too small for rigorous analysis.

---

> ### Author Rebuttal · Authors · 2026-03-31
>
> Dear Reviewer GnZi,
>
> Thank you for taking your valuable time and for recognizing the readability and thoroughness of our empirical analysis. We sincerely appreciate that you found our minimal-intervention approach to be a simple and useful contribution that carries original significance.
>
> ## [W1] Generalization Across Models
>
> We agree that relying exclusively on a single model raises valid questions about generalizability. To confirm our findings' universality , we replicated our full suite of experiments on **Dream 7B Instruct**.
>
> **1. Universality of the Analysis**
>
> Dream's inference dynamics remain highly consistent with that of LLaDA 8B Instruct:
>
> - **Proximity Bias & EOS Dominance:** Premature EOS selection cascades via proximity bias. ([Fig 3(T=128)](https://bit.ly/4lXODal) and [Figure 6(T=32)](https://bit.ly/4s33pOo))
> - **Decisive Initial Trajectory:** Initial positional randomness outperforms continuous temperature sampling ([Fig 4](https://bit.ly/4dkve18)), and initial paths decisively bind final generation ([Fig 5](https://bit.ly/4dfSvBr)).
>
> **2. Planner's Efficacy**
>
> Combining the Planner and EOS Annealing also yields the highest performance on Dream 7B Instruct.
>
> |  | GSM8K | MATH |
> | --- | --- | --- |
> | top1 Prob | 39.9 | 17.2 |
> | **+ Ours** | 52.4 | **21.4** |
> | Prob Margin | 40.9 | 16.2 |
> | **+ Ours** | **54.7** | 19.6 |
> | Ancestral | 28.7 | 12.8 |
> | Temperature | 36.1 | 15.0 |
> | Init Position | 43.9 | 15.4 |
>
> ## [W2-1] Task Specificity & Applicability Scope
>
> While we agree that our planner currently requires model- and task-specific training via verifiable binary rewards, we respectfully view these as deliberate design choices and standard post-training characteristics, rather than fundamental flaws.
>
> **1. Model & Task Specificity:** Training for a specific base model and target task is a natural requirement shared by most alignment paradigms (e.g., RLVR). The planner acts as a tailored alignment module guiding a specific model's unique unmasking dynamics toward a target distribution. While multi-task joint training is theoretically possible, we utilized task-specific planners to isolate the efficacy of initial trajectory shaping per domain.
>
> **2. Verifiable Tasks & Open-Ended Generation:** We chose tasks with unambiguous binary rewards to rigidly validate our trajectory-shaping hypothesis. However, our planner is inherently reward-agnostic. Extending it to open-ended generation woul require only replacing the exact-match verifier with a continuous reward signal (e.g., Reward Model or LLM-as-a-Judge) to score trajectories during training data generation. We will add this discussion to the Limitations and Future Work section.
>
> ## [W2-2] Compute Overhead of Planner Training
>
> We thank the reviewer for pointing out the need for concrete metrics for compute overhead. By 'lightweight,' we are directly contrasting our minimal optimization overhead with the massive memory and compute required to update an 8B base model in RLVR (e.g., via full fine-tuning or PEFT). The computational profile consists of two asymmetric phases:
>
> **1. Offline Trajectory Generation (Inference Only):** Sampling training data requires an upfront compute investment but is strictly a **forward-pass operation** on a frozen 8B backbone. Without gradient computation or optimizer states for the 8B model, peak memory is drastically lower. This is a one-time, highly parallelizable offline process.
>
> **2. Planner Optimization (Lightweight Training):** The 2-layer Transformer planner has **~5M parameters** (<0.1% of the 8B model). Training converges rapidly (~5 minutes on a single A100 GPU) with a negligible memory footprint.
>
> By isolating the 8B model to inference and restricting backpropagation to a 5M module, we avoid the multi-GPU clusters and extensive compute times of standard model alignment.
>
> ## [W3] Sample Size for Rigorous Analysis
>
> We agree that $S=32$ trajectories may be insufficient to draw rigorous conclusions. To solidify our analysis on the binding nature of initial trajectories, we re-ran the constraint experiments in Section 3.2 with significantly larger statistical power:
>
> - **Total Trajectories Per Problem:** Increased from $S=32$ to **$S=256$**.
> - **Sampled Paths:** Increased from 8 to **32 distinct paths per problem** (16 correct, 16 incorrect).
> - **Post-Intervention Sampling:** For each path, we sampled **8 complete sequences** (totaling $ 32 \times 8 = 256 $ final samples per problem).
>
> The revised Figure 5, generated with this scaled-up dataset, is available via this [anonymous link](https://bit.ly/robustness_test). The core conclusion holds: despite substantial late-stage randomness, final generation outcomes are fundamentally constrained by the trajectory established in the first step. We will replace Figure 5 with this high-confidence version in the revised manuscript.
>
> **Reference**
>
> [1] Ye, Jiacheng, et al. "Dream 7b: Diffusion large language models." (2025).

---

> > ### Author Rebuttal · Reviewer_GnZi · 2026-04-02
> >
> > Thanks for the rebuttal. The new results look good and I believe this paper is in a good shape. I am raising my score to 4.

---

> > > ### Author Response · Authors · 2026-04-04
> > >
> > > We sincerely appreciate your thoughtful follow-up and the insightful comments. We will incorporate our responses to your comments into the revised manuscript.

---

### Official Review · Reviewer_dby5 · 2026-03-11

**Soundness:** 3
**Presentation:** 3
**Significance:** 3
**Originality:** 3
**Overall Recommendation:** 4
**Confidence:** 5

**Summary:**

The paper investigates the failure modes of fully non-autoregressive decoding in discrete diffusion language models. Through empirical analysis, the authors demonstrate that the very first unmasking decision disproportionately dictates the success of the entire generation trajectory. To mitigate this, the authors propose a lightweight planner to optimize the initial unmasking positions and an EOS temperature annealing strategy to suppress premature termination. The method is evaluated on reasoning and planning tasks.

**Compliance With Llm Reviewing Policy:**

Affirmed.

**Final Justification:**

My concerns have been resolved, so I have changed my score to 4. I would recommend accepting the paper, but I would not strongly oppose rejection if the AC and other reviewers lean toward rejecting it.

**Key Questions For Authors:**

1. Can you provide preliminary results or analysis demonstrating that proximity bias and premature EOS dominance are prevalent in other dLLM architectures?
2. How does the proposed method compare against recent SOTA samplers (like Fast-dLLM or EB-Sampler) in terms of both generation quality and computation time?

**Limitations:**

Yes.

**Strengths And Weaknesses:**

**Strengths:**
1. The proposed mitigation methods are directly motivated by the empirical findings.
2. The approach is effective while maintaining minimal computational overhead.

**Weakness:**
1. All experiments and observations are based exclusively on a single model (LLaDA 8B Instruct). This raises significant questions about whether proximity bias is an artifact of this specific model's training/architecture or a universal characteristic of all dLLMs. It is unclear if the proposed planner's efficacy will generalize to other dLLMs (e.g., Dream, LLaDA 2.0).
2. The paper only compares the proposed approach with classical sampling baselines (e.g., Top-1 Confidence, Ancestral, Temperature). There are many recent state-of-the-art samplers, such as Fast-dLLM (Wu et al., 2025) and EB-Sampler (Ben-Hamu et al., 2025), that need to be compared.
3. The paper claims negligible overhead from the 5M-parameter planner but provides no end-to-end latency measurements.
4. There are some typos, e.g., line 204 "selction".

I would be happy to raise the score if my main concerns are addressed.

---

> ### Author Rebuttal · Authors · 2026-03-31
>
> Dear reviewer dby5,
>
> Thank you sincerely for taking your valuable time to provide constructive feedback. We appreciate your acknowledgement of the efficacy and clarity of our method.
>
> ## [W1, Q1] Generalization to Other dLLMs
>
> We agree that relying on a single model raises valid questions about generalization. To address the concern, we replicated our full suite of experiments on **Dream 7B Instruct**.
>
> **1. Universality of the Findings**
>
> Dream's inference dynamics remain highly consistent with that of LLaDA 8B Instruct:
>
> - **Proximity Bias & EOS Dominance:** EOS token dominates the initial prediction and it is propagated via proximity bias, thereby truncating the generation window. This behavior is more severe in high-compute setup, degrading the general performance ([Fig 3(T=128)](https://bit.ly/4lXODal) and [Figure 6(T=32)](https://bit.ly/4s33pOo)).
> - **Decisive Initial Trajectory:** Initial positional randomness outperforms continuous temperature sampling ([Fig 4](https://bit.ly/4dkve18)), and initial paths decisively bind final generation regardless of late-stage stochasticity ([Fig 5](https://bit.ly/4dfSvBr)).
>
> **2. Planner's Efficacy**
>
> Consistent with LLaDA 8B Instruct, combining the Planner and EOS Annealing yields the highest performance on Dream 7B Instruct.
>
> |  | GSM8K | MATH |
> | --- | --- | --- |
> | top1 Prob | 39.9 | 17.2 |
> | **+ Ours** | 52.4 | **21.4** |
> | Prob Margin | 40.9 | 16.2 |
> | **+ Ours** | **54.7** | 19.6 |
> | Ancestral | 28.7 | 12.8 |
> | Temperature | 36.1 | 15.0 |
> | Init Position | 43.9 | 15.4 |
>
> ## [W2, Q2] Comparison with Recent SoTA samplers
>
> We respectfully note two key distinctions. First, these samplers primarily target inference acceleration over generation quality. Second, their optimization for semi-autoregressive setups makes direct comparison in our fully NAR setting challenging. Nevertheless, we adapted them for evaluation as follows:
>
> **Implementation Details**
>
> - **Fast-dLLM:** Since its KV-caching requires semi-autoregressive generation, we deployed only its *Confidence-Aware Parallel Decoding* for our fully NAR setup, testing threshold $\tau \in \{0.5, 0.7, 0.9\}$.
> - **EB-Sampler:** Implemented via pseudo-code ($\gamma \in \{0.001, 0.01, 0.1\}$), it proved highly conservative; even at $\gamma=0.1$, average timesteps exceeded 128.
>
> We use average generation timesteps as an objective, implementation-agnostic latency metric. For a fair comparison, we report configurations yielding average steps closest to our fixed $T=32$ budget.
>
> | Category | Method | GSM8K | MATH | Countdown | Sudoku |
> | --- | --- | --- | --- | --- | --- |
> | **Fixed (T=32)** | top1 Prob | 46.6 | 19.2 | 42.2 | 71.2 |
> |  | **+ Ours** | 56.8 | 22.8 | 43.8 | 67.0 |
> |  | Prob Margin | 47.2 | 19.6 | 41.4 | **71.7** |
> |  | **+ Ours** | **58.6** | **23.0** | 45.3 | 69.5 |
> |  | Ancestral | 42.1 | 13.2 | 22.7 | 17.8 |
> |  | Temperature | 45.0 | 19.4 | 43.4 | 69.9 |
> |  | Init Position | 52.6 | 17.6 | 30.1 | 48.3 |
> | **Adaptive** | Fast-dLLM | 40.2 | 19.0 | 41.8 | 58.2 |
> |  | (average timesteps) | (22.7) | (36.8) | (40.0) | (35.4) |
> |  | EB-sampler | 39.1 | 16.8 | **45.7** | 63.0 |
> |  | (average timesteps) | (133.3) | (139.1) | (152.3) | (127.2) |
>
> **Results and Analysis**
>
> - Strictly adhering to a 32-step budget, our method (Prob Margin + Ours) drastically outperforms Fast-dLLM and EB-Sampler on GSM8K (58.6 vs 40.2 / 39.1) and MATH (23.0 vs 19.0 / 16.8).
> - EB-Sampler requires ~**4x more steps**, yet yields significantly lower performance, confirming that simply taking more steps does not solve spatial collapse in fully NAR regimes.
> - Adaptive token allocation alone cannot rectify a flawed early trajectory. Because our planner intervenes solely at the first step, it is entirely orthogonal and can be combined with adaptive samplers.
>
> ## [W3] Latency measurement of the planner
>
> To substantiate our claim of negligible overhead, we measured the average end-to-end wall-clock latency on LLaDA 8B Instruct(1x A100 GPU, Batch Size=1, L=256, T=32). Our method’s latency (2.54s) is identical to the Top-1 Prob baseline (2.65s), with a minor difference falling within hardware variance. The imperceptible computational overhead stems from two design choices:
> 1. **$O(1)$ Intervention:** The 5M-parameter planner (<0.1% of the 8B backbone) is executed **only once** at the first diffusion step. All subsequent steps revert to standard Top-1 decoding.
> 2. **Zero-Cost Annealing:** EOS temperature annealing is a simple scalar multiplication.
>
> ## [W4] Typos
>
> We sincerely thank the reviewer for careful reading and will thoroughly proofread the entire manuscript.
>
> **Reference**
>
> [1] Ye, Jiacheng, et al. "Dream 7b: Diffusion large language models." (2025).
>
> [2] Wu, Chengyue et al. “Fast-dLLM: Training-free Acceleration of Diffusion LLM by Enabling KV Cache and Parallel Decoding.” (2025).
>
> [3] Ben-Hamu, Heli et al. “Accelerated Sampling from Masked Diffusion Models via Entropy Bounded Unmasking.” (2025).

---

> > ### Author Rebuttal · Reviewer_dby5 · 2026-04-03
> >
> > Thank you for your detailed response. My concerns have been resolved, so I am increasing my score to 4.

---

> > > ### Author Response · Authors · 2026-04-04
> > >
> > > Thank you very much for your thoughtful follow-up. We sincerely appreciate your acknowledgement that our rebuttal has adequately addressed your concerns. We will make sure that your valuable feedbacks to be incorporated into the next revision.

---

### Official Review · Reviewer_qX5u · 2026-03-12

**Soundness:** 3
**Presentation:** 4
**Significance:** 3
**Originality:** 4
**Overall Recommendation:** 4
**Confidence:** 5

**Summary:**

Through extensive experiments, this paper first draws two highly interesting conclusions regarding the sampling process of discrete diffusion models: 1) the position of the first unmasked token is of crucial importance, and 2) the premature appearance of the EOS token during generation is highly likely to result in a suboptimal sampling process. In response to these findings, the authors design simple yet ingenious optimization strategies, which have demonstrated performance improvements across a wide range of experiments.

**Compliance With Llm Reviewing Policy:**

Affirmed.

**Final Justification:**

The rebuttal from the authors addressed my concerns well, so I am keeping my original opinion.

**Key Questions For Authors:**

* Most importantly, the reviewer suspects that the observational experiments may harbor certain model or data biases. For example, the authors note in Fig. 3 that LLaDA-instruct has a high probability of first generating the EOS token at the end of the sequence on the GSM8K test set. This could be attributed to the use of EOS padding for data formatting during model training. Furthermore, because the data sequences in GSM8K are relatively short, the trailing tokens of all training samples are EOS, which inadvertently causes the model to learn biased behavioral patterns from skewed data. Therefore, it is necessary to introduce additional base models and conduct similar observational experiments across a wider variety of datasets.
* Regarding the issue of spatial bias, the authors observe in Fig. 3 that "if the first unmasked token is located at the end, it results in an inverse-autoregressive generation order." Following this logic, would it lead to an autoregressive generation order if "the first unmasked token is located at the beginning?"
* For other concerns, please refer to the Weaknesses section.

If the authors can adequately address these concerns, I would be glad to raise my score.

**Limitations:**

yes

**Strengths And Weaknesses:**

**Strengths:**
* The paper is logically structured, flows smoothly, and is highly readable.
* The observational experiments are thorough and detailed, and the resulting conclusions are quite interesting, providing significant value to related research within the community.
* The proposed sampling optimization strategy is remarkably simple yet effective, having been fully validated across extensive experiments.

**Weaknesses:**
* First, the reviewer acknowledges the interesting nature of the conclusions drawn from the observational experiments. However, since these experiments were conducted exclusively using LLaDA-instruct on the GSM8K dataset, there may be certain model or data biases. Conducting observational experiments with a broader range of models across more diverse datasets would make the findings more convincing.
* The proposed strategy is exceptionally simple and ingenious. However, according to the reviewer's understanding, this strategy is not exclusively applicable to LLaDA, nor is it strictly limited to mathematical reasoning tasks like GSM8K and CD. Introducing other base models and testing them on a wider variety of datasets (e.g., Question Answering, Commonsense Reasoning) would make the experimental results much more solid.

---

> ### Author Rebuttal · Authors · 2026-03-31
>
> Dear reviewer qX5u,
>
> Thank you for recognizing the logical structure, readability, and thoroughness of our empirical analysis. We are highly encouraged that you found our conclusions valuable and our strategy simple yet effective.
>
> ## [W1, Q1] Generalization of the Analysis across Models and Datasets
>
> We agree with the reviewer’s insight for attributing data formatting (e.g., EOS padding) as the cause of the model's structural priors. However, we respectfully clarify that this is a universal characteristic of general instruction tuning, rather than an artifact exclusive to GSM8K. More importantly, our core contribution extends beyond identifying EOS dominance; we reveal that proximity bias causes these early token predictions to decisively lock in the final generation trajectory, such that a flawed path cannot be recovered by late-stage randomness.
>
> To empirically demonstrate that these dynamics generalize across both different architectures and varying sequence lengths, we expanded our analysis to Dream 7B Instruct and to the full suite of datasets.
>
> **1. Across Models: Dream on GSM8K**
>
> Dream's inference dynamics match LLaDA almost identically:
>
> - **Proximity Bias & EOS Dominance:** Premature EOS selection cascades via proximity bias. ([Fig 3(T=128)](https://bit.ly/4lXODal) and [Figure 6(T=32)](https://bit.ly/4s33pOo))
> - **Decisive Initial Trajectory:** Initial positional randomness outperforms continuous temperature sampling ([Fig 4](https://bit.ly/4dkve18)), and initial paths decisively bind final generation ([Fig 5](https://bit.ly/4dfSvBr)).
>
> **2. Across Datasets: LLaDA on MATH, Countdown, and Sudoku**
>
> We extended the analysis to the other main datasets.
>
> - **Proximity Bias & EOS Dominance:** MATH exhibits the same EOS dominance([Fig 3 (MATH)](https://bit.ly/4sHoM9f)). In Countdown and Sudoku, this tendency is weaker due to their strict 1-shot example acting as strong structural priors, as analyzed in Sec 4.4. ([Fig 3 (Countdown)](https://bit.ly/4sNGjwA), [Fig 3 (Sudoku)](https://bit.ly/414LcoN))
> - **Decisive Initial Trajectory:** Initial position sampling consistently yields stronger Pass@k than continuous temperature sampling ([Fig 4 (MATH)](https://bit.ly/4v72Y8S), [Fig 4 (Countdown)](https://bit.ly/4sGnn2C), [Fig 4 (Sudoku)](https://bit.ly/4cg5fqE).) Further, final accuracy remains bound to the initial trajectories despite late-stage randomness([Fig 5 (MATH)](https://bit.ly/4m1Jhep)).
>
> We will include these comprehensive results in the Appendix of our revised manuscript.
>
> ## [W2] Generalization of the Proposed Strategy across Models and Datasets
>
> We agree that demonstrating the generalizability across different models and datasets is crucial for solidifying our experimental results. To verify our strategy is not model-specific or domain-exclusive, we integrated it into **Dream 7B Instruct** (GSM8K/MATH) and LLaDA-8B Instruct on **CommonsenseQA (CSQA)**. Our minimal-intervention approach (Planner and EOS Annealing) consistently yields the highest performance.
>
> | **Model** | **DREAM 7B Instruct** | **DREAM 7B Instruct** | **LLaDA 8B Instruct** |
> | --- | --- | --- | --- |
> | **Dataset** | **GSM8K** | **MATH** | **CSQA** |
> | top1 Prob | 39.9 | 17.2 | 65.0 |
> | **+ Ours** | 52.4 | **21.4** | 68.6 |
> | Prob Margin | 40.9 | 16.2 | 67.9 |
> | **+ Ours** | **54.7** | 19.6 | **71.6** |
> | Ancestral | 28.7 | 12.8 | 71.0 |
> | Temperature | 36.1 | 15.0 | 59.3 |
> | Init Position | 43.9 | 15.4 | 66.7 |
>
> ## [Q2] Question on Autoregressive Generation due to Proximity Bias
>
> We thank the reviewer for this highly incisive inquiry into the interplay between two competing forces: the local proximity bias (which builds confidence near the prompt) and the global EOS dominance. We found that **even when a strong proximity bias is artificially established near the prompt, the global structural prior of the EOS token exerts a stronger influence**, ultimately preventing the autoregressive generation in an unconstrained NAR setup.
>
> To explicitly test this, we conducted a new experiment where we forced the model to unmask the first 32 tokens using block-wise decoding for the first 4 steps out of $T=32$. From step 5 onwards, we reverted to standard confidence-based fully NAR decoding. We provide the result via this anonymous [[Link to the new Fig 6]](https://bit.ly/figure_initialAR). As soon as the blocking is released, the generation immediately jumps to the end of the sequence (with the EOS ratio spiking to ~60%), and the inverse-autoregressive order resumes. This confirms that without continuous physical constraints (e.g., Semi-AR as in Figure 7), the inherent EOS dominance ultimately overrides local proximity bias across diverse compute regimes ($T=128$ in Fig3 and $T=32$ in Fig 6).
>
> **Reference**
>
> [1] Ye, Jiacheng, et al. "Dream 7b: Diffusion large language models." (2025).
>
> [2] Talmor, Alon et al. “CommonsenseQA: A Question Answering Challenge Targeting Commonsense Knowledge.”  (2019).

---

> > ### Author Rebuttal · Reviewer_qX5u · 2026-04-04
> >
> > Thanks for the rebuttal. The rebuttal has addressed my question, and I will maintain my positive score.

---

> > > ### Author Response · Authors · 2026-04-05
> > >
> > > We sincerely appreciate the time you took to review our paper and your continued support. We will ensure that your valuable feedback about how our analysis and proposed method can be generalized is incorporated into the revised manuscript.

---

### Official Review · Reviewer_WXwR · 2026-03-12

**Soundness:** 3
**Presentation:** 3
**Significance:** 3
**Originality:** 3
**Overall Recommendation:** 4
**Confidence:** 3

**Summary:**

The paper investigates why fully non-autoregressive decoding in diffusion language models often fails. It identifies a proximity bias in confidence-based unmasking, where early denoising tends to concentrate on neighboring positions, leading to spatial error propagation. The paper further link this to premature EOS selection that truncates generation for reasoning. To mitigate these issues, they propose guiding the initial unmasking positions with a lightweight planner and applying EOS temperature annealing to lower EOS priority early on. Experiments on reasoning and planning benchmarks show these minimal interventions consistently improve performance over heuristic baselines with negligible overhead.

**Compliance With Llm Reviewing Policy:**

Affirmed.

**Final Justification:**

After considering both the paper and the authors’ rebuttal, my final recommendation is positive. The paper is strong in originality and significance, offering an insightful view of diffusion language models through temporal inference dynamics. Overall, the paper is clear, technically solid, and meaningful.

My main concern was whether the phenomenon reflects a general property of confidence-based fully NAR decoding in diffusion LLMs. The rebuttal addressed this concern. While broader validation across more architectural variants would make the claim stronger, the rebuttal meaningfully resolved my main concern and improved my overall assessment.

**Key Questions For Authors:**

1. In Figure 2, which specific diffusion language model is used to produce the demonstrated phenomenon? More importantly, is this phenomenon expected to be a general property of confidence-based fully NAR decoding in dLLMs, or is it tied to a particular model structure design / mechanism of the chosen backbone? In other words, to what extent does the observed behavior in Figure 2 generalize across different dLLMs or architectural variants? This is my main concern. A clear and well-supported answer (e.g., evidence across multiple DLMs design) would improve my score.

**Limitations:**

Yes

**Strengths And Weaknesses:**

Strengths:
1. The paper provides an insightful analysis of diffusion language models from the perspective of inference dynamics along the temporal axis. The paper identifies decoding issues through the lens of trajectory shaping. Specifically, it highlights how early denoising decisions steer the subsequent generation path, which is a novel and compelling angle.
2. The authors systematically investigate the underlying causes of the observed failure mode in fully non-autoregressive decoding, and the analysis is well-supported by experimental evidence. In particular, the characterization of proximity bias as a tendency to accumulate confidence via local token associations helps explain why confidence-based NAR sampling can become unstable.
3. Based on these findings, the paper conducts solid empirical validation of the proposed minimal-intervention approach. The experiments demonstrate the effectiveness of guiding the early inference steps using a lightweight planner and EOS temperature annealing, without fine-tuning the frozen diffusion backbone.

Weaknesses:
1. See Question 1.

---

> ### Author Rebuttal · Authors · 2026-03-31
>
> Dear reviewer WXwR,
>
> Thank you for taking the time to provide constructive feedback on this paper. We deeply appreciate your recognition of the novel and compelling direction of our spatiotemporal analysis. It is truly encouraging that you found our systematic investigation and minimal-intervention approach to be both rational and effective.
>
> ## [Q1] Generalization of the phenomenon demonstrated in Figure 2 across different dLLMs
>
> We sincerely thank the reviewer for raising this critical question. We agree that demonstrating this phenomenon as a fundamental property of confidence-based non-autoregressive(NAR) decoding is central to our paper's narrative.
>
> Figure 2 was generated using the LLaDA 8B Instruct model (as detailed in Section 3). To address your concern and provide empirical evidence, we conducted extensive additional experiments on another state-of-the-art open-source dLLM: **Dream 7B Instruct**.
>
> It is important to note that LLaDA 8B Instruct and Dream 7B Instruct operate on different training paradigms and noise scheduling. The most distinct difference lies in their starting points and training philosophy: LLaDA 8B Instruct is trained from scratch, whereas Dream 7B Instruct is initialized with the Qwen2.5 model, inheriting the knowledge of the autoregressive model. Furthermore, Dream 7B Instruct employs a unique Context-Adaptive Token-Level Noise Rescheduling strategy during training in contrast to the uniform noise schedule of LLADA 8B Instruct. Despite these differences in their initialization and training strategy, we find that the observed phenomenon is remarkably consistent
>
> Table below presents the performance of Dream 7B Instruct and LLaDA 8B Instruct across varying timesteps, with effective token (non-EOS token) counts reported in parentheses to highlight the EOS dominance. Overall, while the exact inflection point for accuracy varies slightly between the models, increasing the timestep budget invariably reduces the number of effective tokens, ultimately leading to severe generation degradation at high compute budgets (e.g., T=128).
>
> **[Performance of Dream 7B Instruct and LLADA 8B Instruct across varying timesteps]**
>
> | Model | timesteps(T) | GSM8K | MATH | Countdown | Sudoku |
> | --- | --- | --- | --- | --- | --- |
> | **Dream 7B Instruct** | 32 | 39.9 (101.8) | 17.2 (78.6) | 50.0 (121.8) | 7.5 (156.2) |
> |  | 64 | 45.5 (78.9) | 17.0 (35.9) | 50.8 (121.2) | 8.6 (147.5) |
> |  | 128 | 33.9 (47.8) | 17.2 (22.9) | 49.6 (120.4) | 0.0 (5.8) |
> | **LLaDA 8B Instruct** | 32 | 46.6 (157.3) | 19.2 (210.3) | 42.2 (182.9) | 71.2 (230.4) |
> |  | 64 | 32.1 (118.6) | 17.0 (170.1) | 41.4 (137.9) | 68.4 (230.6) |
> |  | 128 | 23.3 (100.7) | 15.2 (161.2) | 41.8 (139.9) | 70.5 (230.5) |
>
> Given that these dynamics consistently appear across distinct models, it is suggested that this behavior is a general property of confidence-based NAR decoding. Rather than stemming from specific architectural choices, we hypothesize that it is driven by universal post-training data formats and the inherent nature of language modeling via the following mechanisms:
>
> - **The Origin of Initial EOS Bias:** During instruction-tuning, dLLMs are trained on data formatted as *[Prompt] + [Answer] + [Padding/EOS]*. At the beginning of the inference, the model is highly confident about the structural prior, whereas it faces extreme uncertainty regarding the semantic content.
> - **The Role of Proximity Bias:** Without explicit structural constraints like semi-autoregressive blocking, the model naturally selects the EOS tokens at the end of the sequence. Because language models naturally accumulate confidence based on local context (proximity bias), unmasking the tail-end EOS tokens triggers a cascading, reverse-autoregressive generation of EOS tokens.
> - **High-Compute Degradation:** In a high-compute setup (e.g., $ T=128 $), the model unmasks very few tokens per step. These limited "unmasking slots" are largely consumed by the highly confident EOS tokens at the end of the sequence. By contrast, in a low-compute setup (e.g., $ T=32 $), the model is forced to unmask a large volume of tokens at once. This broader selection window forces the model to sample valid content tokens near the prompt alongside the EOS tokens, effectively establishing valid semantic anchors early in the process.
>
> We additionally show that the main analysis in Section 3 and the proposed method in Section 4 are applicable to other models and datasets as well. Please kindly refer to our response to Reviewer qX5u for these extended results.
>
> We will add these results to the Appendix of the revised manuscript to strengthen our claims.
>
> **Reference**
>
> [1] Ye, Jiacheng, et al. "Dream 7b: Diffusion large language models." *arXiv preprint arXiv:2508.15487* (2025).

---

> > ### Author Rebuttal · Reviewer_WXwR · 2026-04-03
> >
> > Thank you for your response. The rebuttal has addressed my question, and I will increase my confidence.

---

> > > ### Author Response · Authors · 2026-04-04
> > >
> > > We are glad that our rebuttal helped address your question and sincerely appreciate your continued positive evaluation. We will ensure that the discussion regarding the generalizability of our analysis and proposed method is clearly incorporated into the revised manuscript.

---

### Decision · Program_Chairs · 2026-04-30

**Decision:**

Accept (regular)

**Comment:**

The reviewers generally viewed the paper positively, and the discussion supports acceptance. They found the analysis of fully non-autoregressive decoding dynamics insightful, especially the identification of proximity bias, EOS dominance, and the disproportionate importance of early trajectory decisions. The proposed interventions are lightweight, well motivated by the empirical findings, and consistently improve performance across the evaluated reasoning and planning tasks with minimal overhead. While some reviewers initially questioned generalizability beyond a single model and the task-specific nature of the planner, these concerns were largely addressed in the rebuttal through additional results on Dream and broader validation, leaving the overall impression of a technically solid and meaningful contribution.